# Electron-induced chemistry in microhydrated sulfuric acid clusters

Jozef Lengyel[1,2], Andriy Pysanenko[1], Michal Fárník[1]

[1]J. Heyrovský Institute of Physical Chemistry v.v.i., Czech Academy of Sciences, Dolejškova 3, 18223 Prague, Czech Republic
[2]Institut für Ionenphysik und Angewandte Physik, Universität Innsbruck, Technikerstraße 25, 6020 Innsbruck, Austria

*Correspondence to*: Jozef Lengyel (jozef.lengyel@jh-inst.cas.cz), Michal Fárník (michal.farnik@jh-inst.cas.cz)

**Abstract.** We investigate the mixed sulfuric acid-water clusters in a molecular beam experiment with electron attachment and negative ion mass spectrometry, and complement the experiment by DFT calculations. The microhydration of $(H_2SO_4)_m(H_2O)_n$ clusters is controlled by the expansion conditions, and the electron attachment yields the main cluster ion series $(H_2SO_4)_m(H_2O)_nHSO_4^-$ and $(H_2O)_nH_2SO_4^-$. The mass spectra provide an experimental evidence for the onset of the ionic

dissociation of sulfuric acid and ion-pair $(HSO_4^-\cdots H_3O^+)$ formation in the neutral $H_2SO_4(H_2O)_n$ clusters with $n \geq 5$ water molecules, in excellent agreement with the theoretical predictions. In the clusters with two sulfuric acid molecules $(H_2SO_4)_2(H_2O)_n$ this process starts already with $n \geq 2$ water molecules. The $(H_2SO_4)_m(H_2O)_nHSO_4^-$ clusters are formed after the dissociative electron attachment to the clusters containing the $(HSO_4^-\cdots H_3O^+)$ ion-pair structure, which leads to the electron recombination with the $H_3O^+$ moiety generating $H_2O$ molecule and the H atom dissociation from the cluster. The $(H_2O)_nH_2SO_4^-$

cluster ions point to an efficient *caging* of the H-atom by the surrounding water molecules. The electron energy dependencies exhibit an efficient electron attachment at low electron energies below 3 eV, and no resonances above this energy, for all the measured mass peaks. This shows that in the atmospheric chemistry only the *low-energy electrons* can be efficiently captured by the sulfuric acid-water clusters and converted into the negative ions. Possible atmospheric consequences of the acidic dissociation in the clusters and the electron attachment to the sulfuric acid-water aerosols are discussed.

## 1 Introduction

Gas-phase sulfuric acid ($H_2SO_4$) is a key precursor influencing atmospheric aerosol nucleation (Weber et al., 1997; Weber et al., 1999; Sihto et al., 2006; Kulmala et al., 2007). Efficient uptake of water molecules leads to its hydration and the ionic dissociation (Hanson and Eisele, 2000). Progressive hydration of sulfuric acid increases the probability for proton transfer from the acid to water. Theoretical calculations predict the covalently-bonded $H_2SO_4$ as the global minima for $H_2SO_4(H_2O)_n$ clusters with $n \leq 4$, while for $n = 3$ and 4 the hydrogen-bonded, $H_2SO_4\cdots H_2O$, and ion-pair, $HSO_4^-\cdots H_3O^+$, structures are

energetically very close and both structures coexist for these clusters (Re et al., 1999; Temelso et al., 2012a). Upon further hydration $H_2SO_4(H_2O)_n$ for $n \geq 5$ the ion-pair structures become the global energy minima. The ion formation can significantly increase the nucleation rate due to the long-range dipole-charge interactions between the core ions and the polar molecules (Raes and Janssens, 1985, 1986; Yu and Turco, 2000; Lovejoy et al., 2004). The ions can also be formed by the cluster

ionization by cosmic radiation. The cosmic rays are the principal source of ionization and free electrons in the upper troposphere and lower stratosphere, where the electron collisions can influence the gas-phase chemistry. Typically, the electrons are effectively thermalized to low energies ($\gtrsim 1$ eV) via multiple inelastic collisions (Campbell and Brunger, 2016). At these low energies, they are rapidly captured by abundant molecules, in particular $O_2$. Recently, we have shown that the low-energy electrons can be also efficiently captured by hydrated acid clusters generating negative ions and inducing a cascade of ion-molecule reactions (Lengyel et al., 2017a).

Ion-induced nucleation was invoked to explain the correlation between the global cloudiness and the cosmic rays intensity (Carslaw et al., 2002; Harrison and Carslaw, 2003). The temperature-dependent aerosol chamber measurements on the binary $H_2SO_4$-$H_2O$ nucleation showed that the neutral particle formation is preferred at low temperatures, while ion-induced particle formation dominates at higher temperatures (Duplissy et al., 2016). A significant contribution of the ion-induced nucleation to the sulfuric acid aerosols was observed experimentally using a particle beam under atmospheric conditions (Enghoff et al., 2011). Field measurements indicate that the ions are strongly involved in the atmospheric nucleation events (Hirsikko et al., 2007; Hirsikko et al., 2011). However, the ion-induced nucleation alone could not explain the observed nucleation rates (Kirkby et al., 2011). Therefore, other compounds, such as bases (e.g., ammonia and amines) and organic acids, were proposed to contribute to the sulfuric acid aerosol formation (Zhang et al., 2004; Kirkby et al., 2011; Zhang et al., 2012; Almeida et al., 2013; Kürten et al., 2014; Schobesberger et al., 2015; Kürten et al., 2016).

The gas-phase reactions between thermalized electrons and $H_2SO_4$ molecules proceed rapidly yielding the bisulfate anion $HSO_4^-$ (Adams et al., 1986). However, hydration dramatically changes the nature of the electron driven processes (Lengyel et al., 2017a; Lengyel et al., 2017b) in particular via the strong influence of hydration on ion-pair formation. The surrounding water molecules can also stabilize anions, which are otherwise electronically unstable in the gas-phase (Pluhařová et al., 2012). Therefore the electron induced chemistry in sulfuric acid-water clusters is fundamental to our molecular-level understanding of the cluster ion formation and reactivity of aerosol particles.

Recently, a new method for microhydration of molecules has been developed in our laboratory (Kočišek et al., 2016). It enables the generation of the mixed clusters with water where the cluster composition can be relatively well controlled, e.g., clusters of single molecules solvated by one or a few water molecules could be prepared (Kočišek et al., 2016). By this method, we generate the mixed sulfuric acid-water clusters where the sulfuric acid microhydration is controlled. It ought to be mentioned, that it is difficult to control the particle hydration in aerosol experiments of the condensation chamber type, and often dehydrated sulfuric acid particles are obtained. Thus our experiments offer a unique opportunity to investigate the electron induced processes in the binary sulfuric acid-water systems.

We investigate the low-energy electron attachment to $(H_2SO_4)_m(H_2O)_n$ clusters in a crossed-beam experiment where the cluster beam in ultrahigh vacuum is crossed by an electron beam of well-defined adjustable low energies. The formed negative ions are monitored via a time-of-flight (TOF) mass spectrometry. In addition, the small neutral complexes $(H_2SO_4)_m(H_2O)_n$ are characterized by DFT calculations, and the energetics of the initial dissociation step is calculated.

## 2 Experimental and theoretical methods

The experiments were performed on the CLUster Beam (CLUB) apparatus in Prague (Kočišek et al., 2016; Lengyel et al., 2017a). The mixed $(H_2SO_4)_m(H_2O)_n$ clusters were generated in a home-build source via continuous supersonic expansion of the sulfuric acid/water vapor in helium buffer gas through a divergent conical nozzle (100 μm diameter, 2 mm long, and ~30° full opening angle) into the vacuum. The clustering conditions were controlled by heating the $H_2SO_4$ solution in reservoir and by the stagnation pressure of the buffer gas. Reservoir temperatures of 180 °C and buffer gas pressures between 1 and 2 bar were employed. The nozzle was heated independently to a higher temperature (185-190°C) to avoid any condensation. To further increase the water content in the vapor, the humidification of the He buffer gas was performed using a commercially available Elemental Scientific Pergo gas humidifier, where the He-gas passes through a Nafion tube submerged in water reservoir kept at a constant temperature (25°C in the present experiments) and water permeates through the tube and humidifies the buffer gas (Kočišek et al., 2016).

The cluster beam was skimmed and passed through three differentially pumped chambers before entering the ion source of the perpendicularly mounted reflectron TOF mass spectrometer. The voltages on the extraction plates were set for the negative ion detection. The clusters were ionized in the extraction region of the spectrometer using pulsed electron gun at 8 kHz frequency with pulse duration of 2 μs. The electron energy was scanned from 0 to10 eV in 0.2 eV steps and the mass spectra were recorded at each step. The mass spectra presented here were integrated over the entire energy range, since there were no significant qualitative differences in the spectra measured at different electron energies. The extraction pulse to accelerate the negative ions was applied with 0.5 μs delay after the electron pulse to exclude effects of any free electrons in the ionization volume. The ions were detected on the Photonics MCP detector in the Chevron configuration. The electron-energy scale was calibrated using the 4.4 eV and 8.2 eV resonances in the $O^-$ production from $CO_2$ molecules (Denifl et al., 2010). The energy spread of the electron beam was approximately 600 meV. Test measurements with $SF_6$ revealed that below 1.5 eV the electron current passing the ionization region quickly drops, and the current recorded on the Faraday cup does not reflect the actual current in the ionization region. This is caused by the construction of the electron gun which was optimized for operation at energies of several tens of electronvolts for positive ionization. The ion yields below the 1.5 eV can be biased when normalized on the electron current, and large error bars are expected at these low electron energies. The ion yield curves were therefore plotted just at energies above 1.5 eV.

Our experimental observations were supported with quantum chemistry calculations. The calculations were performed for the following clusters: $(H_2SO_4)_m(H_2O)_n$, $m = 1$-2, $n = 0$-5, $(H_2SO_4)_m(H_2O)_nHSO_4^-$, $m = 0$-1, $n = 0$-5 to describe the thermochemistry of the experimentally observed dissociative electron attachment process in which an H-atom left the cluster. We used previously observed energetic minima from literature as the initial structures (Zatula et al., 2011; Husar et al., 2012; Temelso et al., 2012a; Temelso et al., 2012b; Henschel et al., 2014) and equilibrated them in molecular dynamics runs to find the most stable isomers. Molecular dynamics was run on the BLYP/6-31+g* potential energy surface, nuclei were propagated according to classical equations of motion; the constant temperature of 298 K was maintained with Nosé-Hoover

thermostat. The total length of the simulations was 5 ps, with a time step of ~1 fs (i.e. 5000 steps). From the trajectory, several structures were taken and re-optimized at the M06-2X/aug-cc-pVDZ level of theory including zero-point-energy corrections (all structures represented local minima). Altogether 8 different isomers on average were optimized from various initial structures for each cluster type ($(H_2SO_4)_{1-2}(H_2O)_{0-5}$, $(H_2SO_4)_{0-1}(H_2O)_{0-5}HSO_4^-$) including the hydrogen-bonded, $H_2SO_4 \cdots H_2O$,

and ion-pair, $HSO_4^- \cdots H_3O^+$, structures in neutral clusters. Only the most stable isomers were considered for further calculations. We chose the M06-2X functional because of its performance for related systems (Walker et al., 2013; Lengyel et al., 2017a). Localized structures, including hydrogen-bonded, $H_2SO_4 \cdots H_2O$, and ion-pair, $HSO_4^- \cdots H_3O^+$, structures, are in excellent agreement with literature (Re et al., 1999; Henschel et al., 2014). All calculations were performed with the Gaussian 09 software package (Frisch et al., 2013), ABIN code was used for molecular dynamics (Hollas et al., 2015).

**3 Results and discussion**

To investigate the formation of mixed sulfuric acid/water clusters, different concentrations of water and sulfuric acid were exploited in the expansions. Fig. 1 shows the negative ion mass spectra for three different water concentrations. Essentially, the same cluster ions of the types $(H_2SO_4)_m(H_2O)_n HSO_4^-$ and $(H_2O)_n H_2SO_4^-$ could be identified in all three mass spectra (and in other spectra recorded under different expansion conditions, not shown here). However, their relative intensities vary

significantly, reflecting the changes in the neutral cluster composition with water concentration in the expansions. At low water concentrations (molar fraction, $x_{H_2O} \leq 1.5 \times 10^{-3}$; Fig. 1a), the mass spectrum is dominated by anhydrous $(H_2SO_4)_m HSO_4^-$ ($m$ = 0-7) series. The hydrated analogues $(H_2SO_4)_m(H_2O)_n HSO_4^-$ are less populated and observed only up to $m$ = 3. Apart from these series, there is the minor series $(H_2O)_n H_2SO_4^-$. It ought to be mentioned that an uptake of $H_2SO_4$ molecule by $H_2SO_4/H_2O$ clusters is energetically more efficient than an addition of $H_2O$ molecules (Table 1). The efficient sticking results from $H_2SO_4$

forming multiple hydrogen bonds in mixed clusters, as revealed by vibrational spectroscopy (Yacovitch et al., 2013). Upon hydration, the free energy of the addition of a single sulfuric acid molecule to the hydrated cluster is gradually increasing (1), while the free energy for the addition of a single water molecule is essentially decreasing (2). Probably saturation of sulfuric acid by the surrounding water molecules causes this trend. Nevertheless, the negative values of the free energies still favor the nucleation. Our calculated values are in a good agreement with previously published data (Kurtén et al., 2007; Loukonen et

al., 2010; Henschel et al., 2014), which is shown in Table S2.

$$H_2SO_4(H_2O)_n + H_2SO_4 \rightarrow (H_2SO_4)_2(H_2O)_n, \tag{1}$$

$$H_2SO_4(H_2O)_n + H_2O \rightarrow H_2SO_4(H_2O)_{n+1}. \tag{2}$$

**Table 1: Free energies (in kJ mol$^{-1}$, at $T$=298K and $p^0$=1atm) of binary nucleation of $H_2SO_4$ (1) and $H_2O$ (2) to small $H_2SO_4(H_2O)_n$**
**clusters calculated at the M06-2X/aug-cc-pVDZ level of theory.**

|  | $H_2O$ | $H_2SO_4$ |
| --- | --- | --- |

| | | |
|---|---|---|
| H$_2$SO$_4$ | –11.2 | –26.3 |
| H$_2$SO$_4$(H$_2$O) | –7.2 | –31.4 |
| H$_2$SO$_4$(H$_2$O)$_2$ | –3.4 | –32.6 |
| H$_2$SO$_4$(H$_2$O)$_3$ | –9.7 | –45.8 |
| H$_2$SO$_4$(H$_2$O)$_4$ | –4.8 | –38.3 |

With increasing water vapor pressure (Fig. 1b, $x_{H_2O}$ = 2.5×10$^{-3}$), the larger clusters with $m$ > 3 from the (H$_2$SO$_4$)$_m$HSO$_4^-$ series disappear and the mass spectrum contains (H$_2$SO$_4$)$_m$(H$_2$O)$_n$HSO$_4^-$ ions with $m$ = 0-3 and $n$ = 0-4. The series (H$_2$O)$_n$H$_2$SO$_4^-$ increased in relative intensity compared to the spectrum in Fig. 1a. Further, the water vapor concentration was increased to $x_{H_2O}$ = 2.0×10$^{-2}$ by the helium buffer gas humidification. The mass spectrum corresponding to this increase of water content by almost an order of magnitude is shown in Fig. 1c. It is dominated by the hydrated (H$_2$SO$_4$)$_m$(H$_2$O)$_n$HSO$_4^-$ ($m$ = 0-3, $n$ =0-8) cluster ions. Qualitatively, all the spectra exhibit essentially the same major series, which differ only by their relative intensities. Therefore we will analyze the last spectrum containing the most of the mixed water-sulfuric acid species.

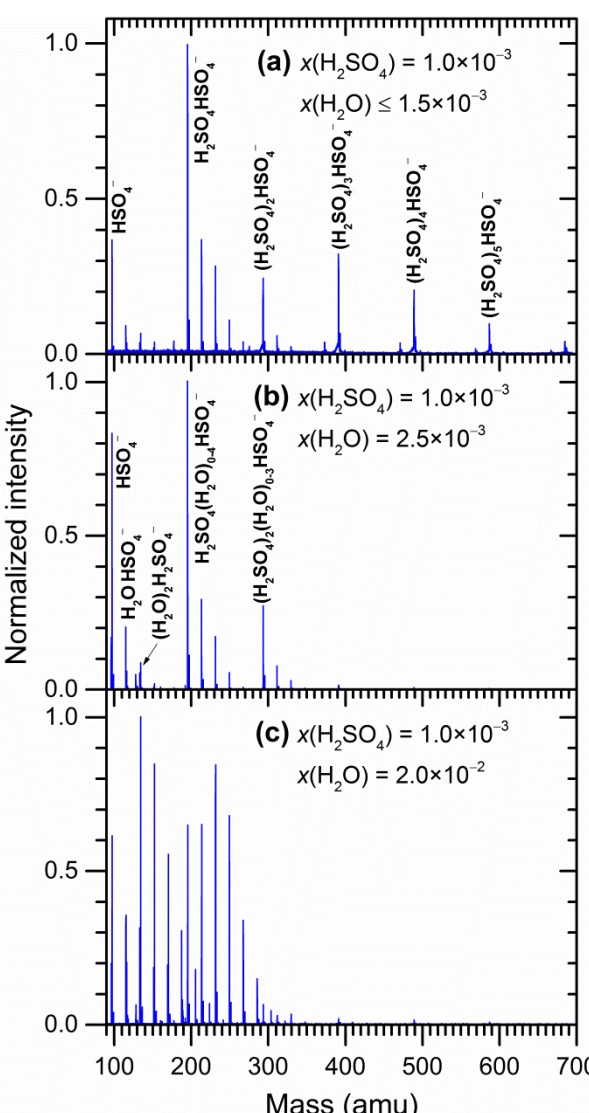

**Figure 1: Negative ion mass spectra integrated in the electron energy range 0-10 eV. Panels (a)-(c) show a decreasing H₂O mole fraction in the vapor coexpanding with constant H₂SO₄ mole fraction in helium. See Figure 2 for analysis of the spectrum (c).**

Fig. 2 shows an analysis of the mass spectrum in Fig. 1c). There are three pronounced cluster ion series: $(H_2O)_nH_2SO_4^-$ ($n = 0$-6), $(H_2O)_nHSO_4^-$ ($n =0$-8) and $H_2SO_4(H_2O)_nHSO_4^-$ ($n =0$-8). In addition, there are also weak $(H_2SO_4)_m(H_2O)_nHSO_4^-$ series with $m = 2$ and 3 discernible in the mass spectrum (the $m = 2$ series is indicated in Fig. 2). It is important to realize that the mass spectra reflect fingerprints of two different processes separated in space and time: (*i*) the neutral cluster generation in the supersonic expansion in the nozzle, and (*ii*) the cluster ionization via electron attachment in the mass spectrometer ionization region. Keeping this in mind, we can start analysing the mass spectra in Fig. 2.

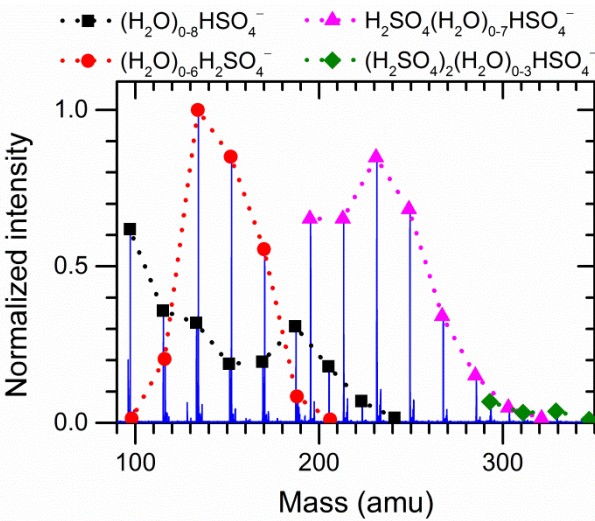

**Figure 2: Negative ion mass spectrum of humidified He/H₂SO₄ ($x=1.0\times10^{-3}$)/H₂O ($x=2.0\times10^{-2}$) gas mixture.**

First, we consider the major $(H_2O)_nH_2SO_4^-$ series. The electron attachment to the pure water clusters is inefficient (Knapp et al., 1987; Lengyel et al., 2017a) thus we can see only the electron attachment to the mixed clusters. The ionization of an isolated sulfuric acid (Adams et al., 1986) proceeds via dissociative electron attachment (DEA):

$$H_2SO_4 + e^- \rightarrow HSO_4^- + H, \tag{3}$$

yielding the relatively strong $HSO_4^-$ ion peak and the $(H_2O)_nHSO_4^-$ series. Nevertheless, we observe also the $(H_2O)_nH_2SO_4^-$ series with the covalently-bonded sulfuric acid molecule. This can be due to the H-atom caging by the water molecules after the DEA process (3), which has been described in our recent paper (Kočišek et al., 2016). In the present case of the DEA process (3) in the cluster, the departing H-atom meets the $H_2O$ molecule and bounces back to remain within the $(H_2O)_nH_2SO_4^-$ complex:

$$H_2SO_4(H_2O)_n + e^- \rightarrow [(H_2O)_nH\cdots HSO_4^-] \rightarrow (H_2O)_nH_2SO_4^-. \tag{4}$$

The calculated structures of $H_2SO_4(H_2O)_n$ clusters, Fig. 3, suggest that from $n = 2$ both acid hydrogens are shielded by the $H_2O$ molecules from the direct departure after the DEA process, while in case $n = 1$ one hydrogen points to the water molecule and the other one can freely leave the cluster. Therefore the intensity of $(H_2O)_nH_2SO_4^-$ series increases steeply from $n = 1$ to 2. The decrease of intensity for $n \geq 3$ is probably due to the neutral $H_2SO_4(H_2O)_n$ cluster size distribution which has an exponentially decreasing character with $n$, typical for small clusters in general (Lengyel et al., 2012; Lengyel et al., 2014). The caging might be also accompanied by subsequent evaporation of water molecule(s). Such evaporative process could contribute to the mass peak at $m/z = 98$ generating $H_2SO_4^-$ ion. However, the negligibly small intensity of this peak (less than 3% of the peak at $m/z = 97$) suggests insignificant contribution of the evaporative processes.

The DEA (3) can lead also to the processes where the H-atom leaves the clusters yielding the $(H_2O)_nHSO_4^-$ series. This series exhibits interesting dependence on $n$ with a secondary maximum at $n = 5$. We assign this maximum to the ion-pair generation in the neutral clusters. According to the theoretical calculations, the molecular structures of sulfuric acid in the

clusters for $n \leq 4$ either correspond to the global energy minima or are energetically very close to the ion-pair structures (Re et al., 1999). Thus, for the small clusters with covalently bound $H_2SO_4$ molecule, we assume the DEA process with hydrogen departure from the clusters:

$$H_2SO_4(H_2O)_n + e^- \rightarrow [(H_2O)_nH \cdots HSO_4^-] \rightarrow (H_2O)_nHSO_4^- + H, \tag{5}$$

The intensities of the corresponding $(H_2O)_nHSO_4^-$ mass peaks then reflect the neutral $H_2SO_4(H_2O)_n$ cluster size distribution, decreasing with the size as outlined above. However, upon hydration with sufficient number of water molecules, the $HSO_3O$–H becomes polarized and heterolytic dissociation occurs yielding the ion-pair $H_3O^+(H_2O)_{n-1}HSO_4^-$ structure. The global minimum for ion-pair structure was reported for $H_2SO_4(H_2O)_n$ clusters with $n \geq 5$ (Re et al., 1999). The electron attachment to these clusters with ion-pair has different character for two reasons. First, the incoming free electron can interact with the $H_3O^+$

moiety or with the dipole of the ion-pair structure of the cluster increasing the electron attachment cross section (Fabrikant et al., 2017). Second, the attached electron will most likely recombine with the $H_3O^+$ in the cluster generating $H_2O$ and H atom. The H-atom can subsequently depart from the cluster, carrying away the excess recombination energy, leaving behind the $(H_2O)_nHSO_4^-$ ion:

$$H_3O^+(H_2O)_{n-1}HSO_4^- + e^- \rightarrow (H_2O)_nHSO_4^- + H. \tag{6}$$

Photodissociation studies of mixed water-hydrogen halide clusters with similar ion-pair structures demonstrated that the dissociating H-atom exits from the clusters efficiently (Poterya et al., 2007; Ončák et al., 2011; Poterya et al., 2014). Therefore it is plausible to assume, that in the present case the H-atom leaves the cluster after the recombination, and carries away the excess energy without any need for further water evaporation from the cluster. Thus the increase of the electron attachment cross section at $n = 5$ due to the ion-pair structure of the clusters leads to the increase of the ion intensity for $(H_2O)_5HSO_4^-$.

The further intensity decrease for $n > 5$ is again due to the decreasing neutral cluster size distribution.

      Finally, there is the $H_2SO_4(H_2O)_nHSO_4^-$ series with the intensity maximum at $n = 2$. This is again consistent with the ionic dissociation of the acid in the clusters. Shields and co-workers (Temelso et al., 2012b) observed the ion-pair formation for clusters containing two sulfuric acid molecules, in which one sulfuric acid molecule is ionically dissociated already in the presence of two water molecules. Thus the maximum at $H_2SO_4(H_2O)_2HSO_4^-$ is in agreement with the ion-pair structure of the

neutral $(H_2SO_4)_2(H_2O)_2$ precursor (i.e. $H_3O^+(H_2SO_4)(H_2O)HSO_4^-$) and with the model outlined above, Eq. (6).

      Thus the observed intensity maxima for $(H_2O)_nHSO_4^-$ and $H_2SO_4(H_2O)_nHSO_4^-$ cluster series can be induced by the chemical reactions in the corresponding neutral clusters. The agreement with theoretical predictions (Re et al., 1999; Temelso et al., 2012a; Temelso et al., 2012b) suggests that the acidic dissociation occurs. To our best knowledge, the $H_2SO_4$ dissociation has not been studied in the mixed water-sulfuric acid clusters experimentally, yet. Therefore our study represents the first

experimental observation which confirms the theoretical calculations of how many water molecules are necessary to induce the ionic dissociation of $H_2SO_4$ molecules.

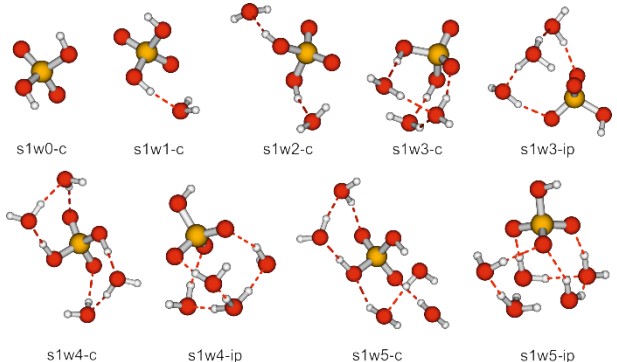

**Figure 3: The observed most stable energy isomers of H₂SO₄(H₂O)ₙ (n= 0-5) clusters with both hydrogen-bonded, H₂SO₄···H₂O, (c) and ion-pair, HSO₄⁻···H₃O⁺, (ip) structures optimized at the M06-2X/aug-cc-pVDZ level of theory.**

Since the mass spectra were measured at different electron energies between up to 10 eV in 0.2 eV steps, the ion-yield curves could be constructed for all the mass peaks in the spectra. However, they were essentially the same for all the ions as exemplified by a few ion yield curves in Fig. 4. The negative ion formation is observed at low electron energies up to 3 eV, which is independent of the composition and degree of hydration of all measured ions. As outlined in the experimental section, we could not determine the position of the ion yield maximum exactly due to the larger error bars on our measurements at the energies below approximately 1.5 eV. Nevertheless, the important point for atmospheric chemistry is that only the secondary electrons with low energy below 3 eV can be directly attached to the mixed sulfuric acid-water clusters. There are no resonant contributions at the higher energies above 3 eV.

The present attachment of low-energy electrons to hydrated sulfuric acid clusters can be compared to other atmospherically abundant clusters, namely the nitric acid hydrates. Those also exhibited a relatively high cross section for electron attachment at low energies measured in range of 0-10 eV. The DEA to HNO₃/H₂O clusters induced three different dissociation channels, namely NO₃⁻, NO₂⁻, and OH⁻ formation. The opening and closing of these reaction channels were specific to cluster size, composition, and degree of hydration (Lengyel et al., 2017a). In contrast to HNO₃, the dissociation of hydrated H₂SO₄ clusters leads exclusively to HSO₄⁻ formation. However, some similarity in the electron attachment can be found between these two cases, namely the H₃O⁺ + e⁻ recombination reaction of ion-pair and the caging of the dissociating molecules by the surrounding cluster environment. As already mentioned, the ion-pair formation efficiently increases the cross section for electron attachment and the electron attachment cross sections are very sensitive to the electron energies (Fabrikant et al., 2017; Lengyel et al., 2017a). Therefore, even if there are only few electrons available for the electron attachment reactions with the cluster particles in atmosphere, the interaction can be very efficiently enhanced by the high cross section, and these reactions may actually contribute to the total budget of the atmospheric HSO₄⁻.

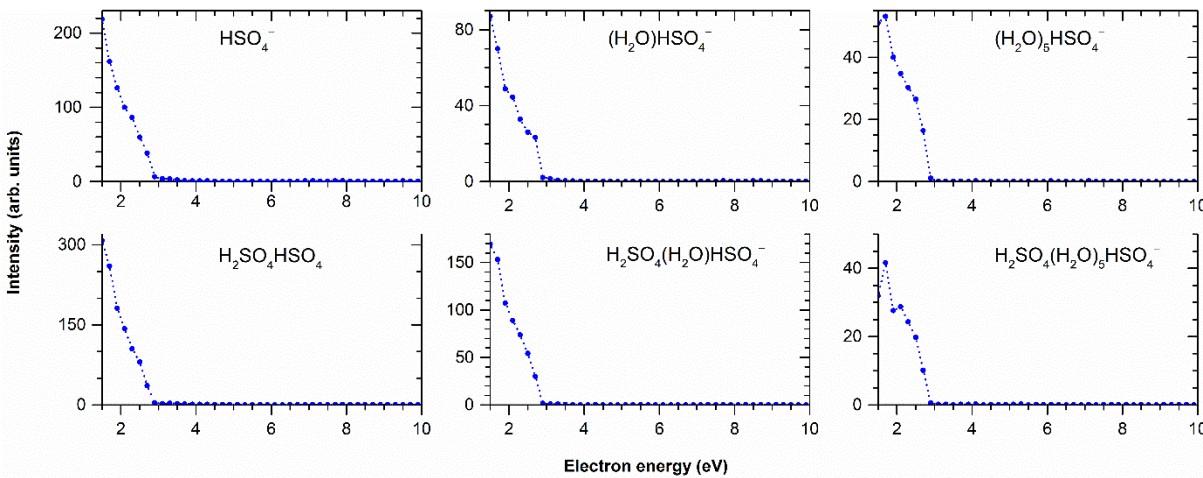

**Figure 4: Ion-yield curves for ionic fragments of $(H_2O)_n HSO_4^-$ and $H_2SO_4(H_2O)_n HSO_4^-$ with different degree of hydration.**

To support the experimental observation, the energetics for the reactions (5) and (6) was evaluated at the M06-2X/aug-cc-pVDZ level and both hydrogen-bonded, $H_2SO_4 \cdots H_2O$, and ion-pair, $HSO_4^- \cdots H_3O^+$, structures were considered in our calculations. Note that the breaking of a specific bond upon the electron attachment primarily depends on the direction of the gradient of the anion potential energy hypersurface at the initial structure, and does not necessarily depend on the asymptotic energetics (Fabrikant et al., 2017). However, the overall thermochemistry gives an overview of how much energy could be released upon the reaction. Our calculations in Fig. 5 show that even the reaction of bare $H_2SO_4$ with free electron yielding bisulfate anion (blue dots) releases the energy of about 8.6 kJ mol$^{-1}$. Upon the hydration of $H_2SO_4$, the energy release gradually increases with the number of water molecules due to the interaction of $HSO_4^-$ with water. Similar behavior was found in the low-energy electron collision with the nitric acid hydrates (Lengyel et al., 2017a). Thus our calculations clearly show that the reactions (5) and (6) above can be driven energetically in the clusters.

Sulfuric acid dimer forms $H_2SO_4 HSO_4^-$ upon the electron attachment quite efficiently with the energy release of 132.2 kJ mol$^{-1}$ (green dots) due to the higher binding energy of $H_2SO_4 HSO_4^-$ compared to $(H_2SO_4)_2$ dimer. With an increasing degree of hydration the energy release somewhat decreases, yet the reaction retains its exothermic character. Thus the reactions (5) and (6) in the clusters with more than one sulfuric acid can also be driven energetically.

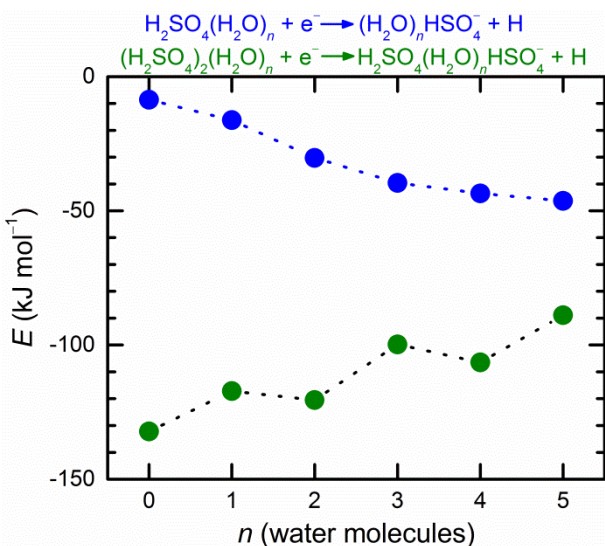

**Figure 5: Reaction energies for the HSO₄⁻ dissociation channels after electron attachment to H₂SO₄/H₂O clusters optimized at the M06-2X/aug-cc-pVDZ level of theory.**

## 4 Conclusion

We have investigated mixed sulfuric acid-water clusters in a molecular beam experiment where the cluster beam was crossed with an electron beam at different energies and electron attachment to the clusters was probed with the mass spectrometry of the negatively charged cluster ions. By the precise control of the sulfuric acid reservoir temperature and the buffer gas humidification we were able to produce the mixed clusters with *different water content*. In all the spectra the main cluster ion series appear: $(H_2SO_4)_m(H_2O)_nHSO_4^-$ and $(H_2O)_nH_2SO_4^-$. Their relative intensities depend on the degree of humidification

By the analysis of these series as a function of the number of surrounding water molecules, we have obtained experimental evidence for sulfuric acid ionic dissociation to form the ion-pairs in the clusters. This occurs in neutral $H_2SO_4(H_2O)_n$ clusters with $n \geq 5$ water molecules, in excellent agreement with the theoretical predictions. Similarly, in the clusters with two sulfuric acid molecules $(H_2SO_4)_2(H_2O)_n$ this process seems to start already with $n \geq 2$ water molecules again in agreement with theory. We introduce the first experimental measurements of the sulfuric acid ionic dissociation in dependence on the controlled stepwise microhydration in the mixed clusters. In relevance to the atmospheric processes the acidic dissociation in small clusters will lead to the increase of the probability that the cluster will capture further molecules from the ambient air. The presence of charge in the cluster containing the ion-pair can increase the interaction between the cluster and the polar (or polarizable) molecules.

Our theoretical investigations of the structure and energetics support the experimental results in several ways: The ion-pair structures are obtained as the energy minima at the degrees of hydration which correspond to the experimental observations. These structures exhibit solvent-separated ion-pairs with the $H_3O^+$ moiety at the cluster exterior. This gives an opportunity for the H-atom to leave the cluster after the electron attachment to these clusters and subsequent $e^- + H_3O^+$

recombination, yielding the experimentally observed clusters with bisulfate anions $HSO_4^-$. For the small $H_2SO_4(H_2O)_n$ clusters with the covalently bound sulfuric acid, the calculated structures suggest the possibility of an efficient H-atom *caging* after the dissociative electron attachment, which is demonstrated experimentally by the presence of $(H_2O)_nH_2SO_4^-$ cluster ions. The calculations also provide support for the proposed reactions (5) and (6) which can be driven energetically in the clusters.

5       Finally, the energy dependencies of the measured mass peaks exhibit an efficient electron attachment at low electron energies below 3 eV, and there are no resonances above 3 eV. Therefore in the atmospheric chemistry only the low-energy electrons below 3 eV can be efficiently captured by sulfuric acid-water aerosols and converted into the negatively charged ions. It is also interesting to note, that the electron attachment efficiencies qualitatively exhibit the same energy dependencies for all the species independently of the degree of hydration (at least for electron energies above 1.5 eV).

*Data availability.* All of the presented data are available from the corresponding authors upon request.

  *Acknowledgements.* This research was funded by the Czech Science Foundation, grant no. 17-04068S. JL acknowledges the support through the Lise-Meitner Programme of the Austrian Science Fund (FWF), project no. M1983-N34.

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
