# Peer review of "Electron-induced chemistry in microhydrated sulfuric acid clusters"

_Atmospheric Chemistry and Physics, 2017_

## Referee Comment (RC1) · Anonymous Referee #1 · 18 Aug 2017

Lengyel, Pusanenko and Fárnik have studied the attachment of electrons to small sulfuric acid - water clusters, and complement their experimental results by density functional theory calculations. I can not comment in depth on the experimental details of the study (as far as I can tell they seem sound, but hopefully another reviewer will have more expertise on this), and will thus focus my review mainly on the atmospheric chemical and computational aspects.

Sulfuric acid, as stated by the authors, is one of the most important molecules for atmospheric aerosol processes, in particular new-particle formation. While neutral pathways are likely to dominate new-particle formation in the contemporary boundary layer, (as shown e.g. by results from the CLOUD consortium), ion-mediated processes can be an important minor channel, and also increase in importance with altitude (as well as

being more important in cleaner atmospheres, including the preindustrial and hopefully also the future). The present study is thus likely to be of interest to the atmospheric science community.

The computational methods employed here are fairly modest - state-of-the-art studies on similar systems nowadays tend to employ coupled-cluster energy corrections to account for the limitations of DFT, or at the very least use triple-zeta basis sets in the DFT calculations of final energies (rather than double-zeta as done here). However, the methods used are of sufficient quality for the required purposes (i.e. capturing reaction energy differences on the order of 0.1 eV or more). The computations are also, for understandable technical & computational reasons, also restricted only to closed-shell species (plus an isolated H radical), but then quite extensive conclusions are drawn about the structures and behaviour of radical (transient or final) structures, e.g. concerning "H caging" and so on.

I have one major question for the authors, along with one suggestion for (a fairly limited set of) additional computations, and a number of minor suggestions or corrections.

My main question is, why does the concentration of clusters with 3 or more H2SO4 molecules drop when more water is added (e.g. figure 1)? According to both classical thermodynamics and quantum chemical calculations (as well as chemical common sense), water should promote clustering of sulfuric acid significantly. Thus one would assume that the concentration of larger clusters would increase when the water content goes up. Or to put it another way, typically the nucleation rate increases with increasing RH for constant [H2SO4] and T. Does perhaps the absolute H2SO4 concentration (which the authors don't actual report) decrease from the low-water runs to the high-water runs?

My main suggestion is that the authors add some calculations on the (H2O)nH2SO4-radical anionic clusters to support their extensive speculation on "H caging" and similar effects. While I understand their reluctance to work with larger open-shell clusters

(with more than one H2SO4/HSO4- moiety), the (H2O)nH2SO4- system with small (e.g. 1-4) n is certainly treatable at the UM06-2X/aug-cc-pVDZ level, and while the resulting energies may not be as accurate as for the closed-shell systems, the structures would certainly be good enough to investigate the "caging" phenomenon the authors repeatedly speculate about.

Minor comments:

-In the abstract, the authors state that "the (H2SO4)m(H2O)nHSO4− clusters are formed after the dissociative electron attachment to the clusters containing the (HSO4−ÂůÂůÂůH3O+) ion-pair structure". This is a reasonable conclusion to draw from their data, but their reasoning is based on somewhat indirect evidence - I would modify the sentence to account for this, e.g. by adding a word "likely", or starting the sentence with "Our results indicate that" or something similar.

-In the introduction, the authors call the sulfuric acid - water clusters where the sulfuric acid remains undissociated "neutral". While this is not wrong, it can lead to confusion, as also the ion-pair clusters (HSO4-…H3O+) are "neutral" in the sense of having a overall electrical charge of zero. I would thus recommend the authors use some other term to denote the undissociated clusters. (Later on they themselves use the term "covalently-bonded", which is one option; "hydrogen-bonded molecular cluster" would be even more accurate but somewhat lengthy.)

-On page 7, the authors talk about the "presumably larger dipole moment" of the ion-pair structures. They do not need to presume anything about dipole moments, as their quantum chemical calculations contain the dipole moments of all their structurtes - they should instead report (in the supplement) the dipole moments of all their global minima, and for the "borderline" cases where the molecular cluster and ion pair structures are close in energy, perhaps report dipole moments for the best structures of both cases. These data could then be used to see whether the reasoning is indeed correct or not.

-On page 9, the authors mention the "polarization of the second H2SO4 molecule when

the HSO4- ion is generated" as the reason for the very exothermal formation of HSO4-H2SO4 (and neutral free H) from (H2SO4)(H2SO4)-. This is not in itself wrong, but a more informative/illustrative way to phrase it would be that the HSO4-(H2SO4) cluster has a much higher binding energy (by tens of kcal/mol) than the (H2SO4)(H2SO4) cluster.

-Line 2 of section 2: "home-build" should be "home-built".

-Figure caption of figure 1: "decreasing H2O mole fraction" should presumably be increasing (as we go from a to b to c, x(H2O) goes up)

---

## Short Comment (SC1) · 7 Sep 2017

Author's response to the discussion paper:

**Electron-induced chemistry in microhydrated sulfuric acid clusters**

Jozef Lengyel[1,2], Andriy Pysanenko[1], Michal Fárník[1]

[1]*J. Heyrovský Institute of Physical Chemistry v.v.i., Czech Academy of Sciences, Dolejškova 3, 18223 Prague, Czech Republic*
[2]*Institut für Ionenphysik und Angewandte Physik, Universität Innsbruck, Technikerstraße 25, 6020 5 Innsbruck, Austria*

*Correspondence to: Jozef Lengyel (jozef.lengyel@jh-inst.cas.cz), Michal Fárník (michal.farnik@jh-inst.cas.cz)*

**Reply to the interactive comment of Referee #1:**

We would like to thank the referee for his valuable comments and overall positive evaluation of our manuscript. Before addressing his particular points in more detail, we would like to make a general comment.

We would like to stress that the major contribution of our present paper is the experiment. The calculations were performed to provide a support for the experimental conclusions. The major conclusions, e.g., about the acidic dissociation or fragment caging in the clusters could be derived essentially just based on the experimental evidence. Actually, such conclusions were derived previously for similar systems (nitric acid–water clusters) just from the experimental mass spectra in the early work of Castleman's group [Kay, B. D., Hermann, V., and Castleman Jr., A. W.: Studies of gas-phase clusters: The solvation of $HNO_3$ in microscopic aqueous clusters, Chem. Phys. Lett., 80, 469, 1981]: the number of water molecules needed to acidically dissociate an $HNO_3$ molecule in $HNO_3(H_2O)_N$ clusters was derived from the shape of the $HNO_3(H_2O)_nH^+$ mass spectra (see also our work: [Lengyel, J., Pysanenko, A., Kočišek, J., Poterya, V., Pradzynski, C. C., Zeuch, T., Slavíček, P., and Fárník, M.: Nucleation of mixed nitric acid-water ice nanoparticles in molecular beams that starts with a $HNO_3$ molecule, J. Phys. Chem. Lett., 3, 3096, 2012]). Recently, we have shown that for the $(HNO_3)_M(H_2O)_N$ clusters the conclusions drawn from the mass spectra are actually in excellent agreement with the theoretical calculations [Lengyel, J., Ončák, M., Fedor, J., Kočišek, J., Pysanenko, A., Beyer, M. K., and Fárník, M.: Electron-triggered chemistry in $HNO_3/H_2O$ complexes, Phys. Chem. Chem. Phys., 19, 11753, 2017]. Therefore, in the present

case we implement the theoretical calculations to support our conclusions drawn from the experimental evidence –and once again the experiment and theory are in excellent agreement.

We understand that our present level of theory might not exceed the theory level required for a stand-alone theoretical paper. However, that was not our ambition –we performed the calculations at the level accessible to our experimental group and they were in agreement with the experiment. Therefore we published them alongside with the experiment as they can provide more (pictorial) insight into what is actually happening in the clusters. It ought to be mentioned that even the calculations performed at the highest possible level of theory are not guaranteed to deliver a reliable picture of what is going on in the real system, unless they are backed up by some experimental evidence.

Besides, we would like to stress that our benchmark calculations proved that the used computational approach using double zeta basis set are in reasonable agreement with the higher-level *ab initio* methods. **Table 1** summarizes the benchmark calculations of electron affinity of $HSO_4$, ionization potential of $H_2SO_4$, and reaction enthalpies for deprotonation of gas-phase $H_2SO_4$ calculated at different levels of theory. The M06-2X/aug-cc-pVDZ energies are comparable with the CCSD/aug-cc-pVDZ values with the exception of the $IP(H_2SO_4)$. The comparison of double-zeta with triple-zeta basis sets of the M06-2X functional shows that there is essentially constant shift from the experimental values and therefore we do not expect any significant shift in reaction energies even upon hydration. The calculated reaction enthalpies for deprotonation of gas-phase $H_2SO_4$ are in good agreement with the experimental value. The error of the DFT method is 0.1-0.2 eV. Please note that, in the present work, chemical trends with respect to hydration are of the main concern, and a possible systematic shift of few tenths of eV does not influence our conclusions.

**Table 1:** Electron affinity of $HSO_4$, ionization potential of $H_2SO_4$, and enthalpy of deprotonation at various levels of theory (all in eV). DZ and TZ represent aug-cc-pVDZ and aug-cc-pVTZ, respectively. Enthalpies were calculated at 298.15 K within the harmonic approximation.

|  | B3LYP/DZ | M06-2X/DZ | M06-2X/TZ | MP2/DZ | CCSD/DZ | Experiment |
|---|---|---|---|---|---|---|
| $EA(HSO_4)$ | 4.69 | 4.92 | 5.01 | 5.21 | 4.92 | $4.75\pm0.10^{a}$ |
| $IP(H_2SO_4)$ | 11.4 | 11.6 | 11.8 | 12.4 | 12.5 | $12.4\pm0.05^{b}$ |
| $\Delta H(H_2SO_4{\rightarrow}H^{+}{+}HSO_4^{-})$ | 13.7 | 13.5 | 13.5 | 13.4 | 13.6 | $13.4\pm0.24^{a}$ |

$^{a}$ Wang, X.B., Nicholas, J.B., Wang, L.S.: Photoelectron spectroscopy and theoretical calculations of $SO_4^{-}$ and $HSO_4^{-}$: Confirmation of high electron affinities of $SO_4$ and $HSO_4$, J. Phys. Chem. A, 104, 504, 2000.

$^{b}$ Snow, K.B., Thomas, T.F.: Mass spectrum, ionization potential, and appearance potentials for fragment ions of sulfuric acid vapor, Int. J. Mass Spectrom. Ion Processes, 96, 49, 1990.

Now we would like to address the actual referee's points:

I) My main question is, why does the concentration of clusters with 3 or more H2SO4 molecules drop when more water is added (e.g. figure 1)? According to both classical thermodynamics and quantum chemical calculations (as well as chemical common sense), water should promote clustering of sulfuric acid significantly. Thus one would assume that the concentration of larger clusters would increase when the water content goes up. Or to put it another way, typically the nucleation rate increases with increasing RH for constant [H2SO4] and T. Does perhaps the absolute H2SO4 concentration (which the authors don't actually report) decrease from the low-water runs to the high-water runs?

The referee is, indeed, correct that increasing water concentrations promotes the sulfuric acid clustering in condensation chamber type experiments where equilibrium conditions can be reached. However, such conditions are far from our experimental method of the cluster generation. In supersonic expansions the clusters are generated in a very limited space and time span. Typically the molecules undergo ~$10^4$ collisions in the expansion and they all happen within ~20 nozzle radii. In our case this represents ~2 mm from the nozzle throat, and the molecules pass through this region in a few microseconds. After that (freezing/quitting surface) the molecules undergo no more collisions and the clusters which have been formed essentially do not change until the interaction with the electron beam in the mass spectrometer 2.5 m downstream from the nozzle. An important aspect is also the strong cluster cooling in the expansion due to the inelastic collisions with the buffer gas –the buffer gas atoms carry away the energy in their kinetic energies transforming the internal cluster energy into the kinetic energy of the gas flow in the direction of the beam. The clusters can be cooled by this mechanism to very low temperatures (e.g. for pure water clusters temperatures below 100 K can be routinely achieved). Due to the gas rarefaction the supersonic expansion is a non-equilibrium process and the cluster composition is determined by the collisions between the constituent molecules in the expansion and by the cooling in the collisions with the buffer gas.

In the spectra in figure 1 in the main paper, the He buffer gas pressure is kept constant at ~2 bar and we control the $H_2O$: $H_2SO_4$ ratio in the vapor by controlling the temperature of the reservoir $T_R$ containing the sulfuric acid. Our aim was to obtain the mixed clusters. Panel (a) in figure 1 corresponds to concentrated sulfuric acid (98.5%) in the reservoir at $T_R$ = 453 K. The partial vapor pressure of water and sulfuric acid under these conditions was 3.09 mbar and 2.12 mbar, respectively [Perry, R. H., Green D. W., Maloney, J. O.: Perry's chemical engineers' handbook, 7th, MacGraw-Hill, New York, 1997], corresponding to the mole

fractions indicated in the figure. These conditions yielded almost exclusively the pure $H_2SO_4$ clusters. The spectrum in panel (b) corresponded to the sulfuric acid concentration of 98.0% at the same $T_R$, i.e., the partial vapor pressures of $H_2O$ and $H_2SO_4$ were 4.92 mbar and 2.04 mbar, respectively [Perry, R. H., Green D. W., Maloney, J. O.: Perry's chemical engineers' handbook, 7th, MacGraw-Hill, New York, 1997]. In panel (c) we added more water directly into the carrier gas using the humidifier –this is a new very successful method introduced by our group just recently for microhydration of biomolecules [Kočišek, J., Pysanenko, A., Fárník, M., and Fedor, J.: Microhydration prevents fragmentation of uracil and thymine by low-energy electrons, J. Phys. Chem. Lett., 7, 3401, 2016]. This way we increased the partial water vapor pressure to approximately 42 mbar while the partial vapor pressures of $H_2SO_4$ remained 2.04 mbar. This finally yielded the substantial hydration and the mixed clusters.

Thus the water concentration increases from (a) to (c). First, there is very little water and there are mostly collisions between $H_2SO_4$ molecules and He generating the pure $(H_2SO_4)_N$ clusters. Increasing the water concentration, the collisions with water molecules become more frequent and some $H_2O$ molecules stick to the sulfuric acid and the mixed clusters appear. At the same time the collisions between $H_2SO_4$ molecules become less frequent resulting in smaller $(H_2SO_4)_N$ clusters.

Under equilibrium conditions in nucleation chambers, the most likely mechanism how the water promotes clustering is by generating mixed clusters in the first place, and subsequently the water is replaced with $H_2SO_4$ molecules in these clusters. Therefore, essentially only sulfuric acid clusters without water can be observed in the nucleation chamber type experiments. However, this is not the case in our supersonic expansions where the collisions cease after a short time before all water can be replaced in collisions with $H_2SO_4$ molecules. Our aim was to generate and investigate the elusive mixed clusters as the early stage in the sulfuric acid nucleation upon humid conditions. The molecular beams are an ideal tool for such type of experiment. The hypothesis of $H_2SO_4$ replacing water molecules could be also tested in our molecular beam experiment by pickup of $H_2SO_4$ on pure $(H_2O)_N$ clusters, however, it is technically very demanding far beyond our present experiment.

II) My main suggestion is that the authors add some calculations on the (H2O)nH2SO4-radical anionic clusters to support their extensive speculation on "H caging" and similar effects. While I understand their reluctance to work with larger open-shell clusters (with more than one H2SO4/HSO4- moiety), the (H2O)nH2SO4- system with small (e.g. 1-4) n is certainly treatable at the UM06-2X/aug-cc-pVDZ level, and while the resulting energies may not be as accurate as for the closed-shell systems, the structures would certainly be good enough to investigate the "caging" phenomenon the authors repeatedly speculate about.

We followed the suggestion of the referee and calculated the negatively charged $(H_2O)_nH_2SO_4^-$ clusters system with $n = 1-5$. **Fig. 1** represents the most stable energy isomers of $(H_2O)_nH_2SO_4$ ($n= 0-5$) clusters (figure 3 of the main paper) re-optimized as the negative ions at the M06-2X/aug-cc-pVDZ level of theory. Comparing the neutral and corresponding anionic structures, the main difference was reducing a dihedral angle between two OH groups from ~158° to ~75° of the sulfuric acid molecule, which resulted in changing of the water molecule orientation in the mixed clusters. The re-optimization of the s1w5-c structure was always followed by spontaneous $H_3O$ formation.

[Figure]

**Figure 1.** Re-optimized neutral most stable energy isomers of $H_2SO_4(H_2O)_n$ ($n= 0-5$) clusters with both covalent-bonded (c) and ion-pair (ip) structures as anions. The clusters were optimized at the M06-2X/aug-cc-pVDZ level of theory. The corresponding neutral structures are shown in figure 3 of the main paper.

Although, we have performed the calculations as suggested by the referee, we are not convinced that any further conclusions about the hydrogen caging can be made from these calculations. It ought to be mentioned that the caging, in the present case, is rather a solid experimental observation than a speculation. The electron attachment to a molecule is in principle a dissociative process (DEA) even if a stable anion exists for the molecule (even for zero kinetic energy electrons) [Fabrikant, I. I., Eden, S., Mason, N. J., and Fedor, J.: Recent progress in dissociative electron attachment, Adv. At. Mol. Opt. Phys., 66, 545, 2017]. For sulfuric acid the $H_2SO_4^-$ ion does not exist –neither experimentally nor theoretically and the hydration does not seem to stabilize the small $(H_2O)_n$ $H_2SO_4^-$ species sufficiently. Yet, the observation of these ions in the mass spectra is an unambiguous experimental fact, therefore the $H_2SO_4 + e^- \rightarrow HSO_4^- + H$ dissociation after the DEA process has to be hindered and the hydrogen must be caged by the solvent in order to observe the $(H_2O)_nH_2SO_4^-$ series in the spectrum. We do not wish to speculate what kind of structural arrangement the $(H_2O)_nH_2SO_4^-$ clusters assume, since there are probably numerous possibilities (there will be also a distribution of the neutral starting $(H_2O)_nH_2SO_4$ cluster configurations of which figure 3 in the main paper represents only some examples to illustrate that there are neutral structures where the water molecules can hinder the free hydrogen dissociation).

**Minor comments:**

1) In the abstract, the authors state that "the (H2SO4)m(H2O)nHSO4- clusters are formed after the dissociative electron attachment to the clusters containing the (HSO4□° u°u°uH3O+) ion-pair structure". This is a reasonable conclusion to draw from their data, but their reasoning is based on somewhat indirect evidence - I would modify the sentence to account for this, e.g. by adding a word "likely", or starting the sentence with "Our results indicate that" or something similar.

We have changed the corresponding sentence according to referee's suggestion. It now starts with "Our results indicate that…"

2) In the introduction, the authors call the sulfuric acid - water clusters where the sulfuric acid remains undissociated "neutral". While this is not wrong, it can lead to confusion, as also the ion-pair clusters (HSO4-: : :H3O+) are "neutral" in the sense of having a overall electrical charge of zero. I would thus recommend the authors use some other term to denote the undissociated clusters. (Later on they themselves use the term "covalently-bonded", which is

one option; "hydrogen-bonded molecular cluster" would be even more accurate but somewhat lengthy.)

We have changed the term "neutral" to "covalently bonded $H_2SO_4$" wherever possible in the main article according to the referee's suggestion.

3) On page 7, the authors talk about the "presumably larger dipole moment" of the ion-pair structures. They do not need to presume anything about dipole moments, as their quantum chemical calculations contain the dipole moments of all their structures – they should instead report (in the supplement) the dipole moments of all their global minima, and for the "borderline" cases where the molecular cluster and ion pair structures are close in energy, perhaps report dipole moments for the best structures of both cases. These data could then be used to see whether the reasoning is indeed correct or not.

The argument that the dipole moment of the cluster increases upon the ion pair generation in the cluster was used for the explanation of the mixed $HNO_3(H_2O)_N$ cluster mass spectra [Kay, B. D., Hermann, V., and Castleman Jr., A. W.: Studies of gas-phase clusters: The solvation of $HNO_3$ in microscopic aqueous clusters, Chem. Phys. Lett., 80, 469, 1981]. Essentially the same argument was used in the interpretation of recent experiments where the mixed $HCl(H_2O)_N$ clusters were deflected in electric fields [Guggemos, N., Slavíček, P., and Kresin, V. V.: Electric dipole moments of nanosolvated acid molecules in water clusters, Phys. Rev. Lett., 114, 43401, 2015] (and also earlier for $HNO_3(H_2O)_N$ cluster [Moro, R., Heinrich, J., and Kresin, V. V.: Electric dipole moments of nitric acid-water complexes measured by cluster beam deflection, AIP Conf. Proc., 1197, 57, 2009]). Although, it ought to be mentioned that in the $HCl(H_2O)_N$ case the theoretical calculations showed that the change in the cluster dipole moment upon the acidic dissociation was relatively small and could be overlapped by dynamic effects. In our present investigation of the mixed $H_2SO_4(H_2O)_N$ clusters, the calculations show very broad range of dipole moments (e.g., from 0.3 D to 4.6 D for $N = 5$ clusters) which depend rather on the cluster structure than on the acidic dissociation (see **Fig. 2** –also added in SI now). Most likely, not only the energy minimum structure but many different cluster structures are generated in the supersonic expansion.

[Figure]

**Figure 2.** Selected local minima of neutral, $H_2SO_4 \cdots H_2O$, (a-d) and ion-pair, $HSO_4^- \cdots H_3O^+$, (e-h) structures in $H_2SO_4(H_2O)_5$ clusters and the corresponding dipole moments.

However, the increase of the peak intensities in mass spectra as a function of the number of water molecules $n$ in the clusters is an unambiguous experimental observation. This increase has to be connected with a change in some physical properties of the clusters with $n$. The obvious property which changes with $n$ is the occurrence of the ion pair from a certain $n$ on –please, see [Kay, B. D., Hermann, V., and Castleman Jr., A. W.: Studies of gas-phase clusters: The solvation of $HNO_3$ in microscopic aqueous clusters, Chem. Phys. Lett., 80, 469, 1981] for more detailed argumentation. In the present case it can be the presence of the $H_3O^+$ in the cluster which leads to the more efficient electron attachment and generation of the negative $(H_2O)_n HSO_4^-$ clusters, rather than a larger dipole moment of the zwitterionic clusters (which does not have to be larger as the present calculations suggest).

Thus, thanks to this referee's comment, we have actually discovered an interesting issue, which might have been not quite correct in the past and recent literature and might require some attention in the future. Therefore we have modified our arguments correspondingly (two sentences: page 7 line 10; page 10 line 17).

4) On page 9, the authors mention the "polarization of the second H2SO4 molecule when the HSO4- ion is generated" as the reason for the very exothermal formation of HSO4- H2SO4 (and neutral free H) from (H2SO4)(H2SO4)-. This is not in itself wrong, but a more

informative/illustrative way to phrase it would be that the HSO4-(H2SO4) cluster has a much higher binding energy (by tens of kcal/mol) than the (H2SO4)(H2SO4) cluster.

We have changed the corresponding sentence according to referee's suggestion.

5) Line 2 of section 2: "home-build" should be "home-built".

The phrase has been corrected.

6) Figure caption of figure 1: "decreasing H2O mole fraction" should presumably be increasing (as we go from a to b to c, x(H2O) goes up)

The referee is correct, we apologize for this confusion and change the corresponding text.

---

## Referee Comment (RC2) · Anonymous Referee #2 · 8 Sep 2017

**Overall:**

The manuscript describes mass spectrometric measurements and quantum chemical calculations on the sulphuric acid water system. The authors propose a caging mechanism in which an intermediate [(H2O)nH.HSO4-] species forms, before the neutral hydrogen atom is released. This mechanism applies probably only in cases where H2SO4 is charged directly by an e-. However, to my knowledge thermal electrons will most likely attach first to more abundant species. H2SO4 would then react with those ions and form HSO4-. If the authors have any references to more recent work that indicate direct charging of H2SO4 by free electrons in the Earth atmosphere they should cite them. H2SO4 is an important compound for the formation of aerosol particles, even if more recent research draws a more complex picture. This aspect of the manuscript

fits therefore well into the scope of ACP. In general I agree with Ref 1 remarks about the quantum chemical calculations and this part should be extended.

Scientific:

- To what temperature does the He/H2SO4/H2O gas mixture drop after nozzle?

- What relative humidity would be reached in the cluster formation zone after the noz-zle?

- How long is the reaction time to form these clusters (before ionisation)?

- How long is the reaction time the charged clusters (after ionisation)?

- Is it possible that some water is condensing on the cluster after the ionisation?

- Could you determine the lifetime of the intermediate by adjusting the reaction time?

- I assume the charged clusters are accelerated by an electric field in a low pressure environment, is evaporation and fragmentation a potential problem in your setup?

- How do your dissociation constants compare to bulk phase sulphuric acid?

Technical:

Data statement is missing. https://www.atmospheric-chemistry-and-physics.net/about/data\_policy.html

A brief overview of current developments in the quantum chemical treatment of H2SO4-H2O clusters is missing.

P 1 L 19: I think aerosols should be replaced with clusters as the paper shows only results for clusters containing a few sulphuric acid molecules

P 1 L 27: I suggest to replace neutral structure by undissociated structure and (optionally) ion pair structure by dissociated structure (ps: probably solved by following referee 1 request) ACPD
P2 L 3-9: Is the Sulphuric acid charging by e- capture actually important for the production of charged sulphuric acid cluster (especially in the context of boundary layer nucleation events)? Some older and more recent literature is missing in the context of ion induced nucleation Raes and Jansen 1985 DOI:10.1016/0021-8502(85)90028-X (or even earlier), a more recent one would be Kirkby et al 2011 (DOI: 10.1038/nature10343) or the most recent for H2SO4 water Kürten et al 2016 (10.1002/2015JD023908). There are also field observations that try to estimate the fraction of ion induced nucleation in the total nucleation rate (Hirsiko et al 2011, doi:10.5194/acp-11-767-2011).

P2 L14-15: Charged sulphuric acid clusters in the atmosphere where already measured before, example literature Eisele, 1989 DOI: 10.1029/JD094iD02p02183 I have the impression this sentence is out of context and should be moved to the previous paragraph.

Methods: P4 L 2: What was the criterion for choosing the clusters for re-optimisation? How many isomers were used?

Results: P4 L 17: The results of the quantum chemical calculations should be compared to previous work.

P8 L 21: could this be quantified more? Is the cross section and e- concentration high enough so that electron attachment in the troposphere is a major source of HSO4-? What is the free e- concentration in the troposphere?

P9 L 3: How was that calculated? You probably used all structures for a given isomer, please be a bit more precise here.

Conclusion: P10 L 16-17: The dipole in the dissociated cluster could also enhance the uptake of ions such as NO3-. With the quantum chemical calculations you could calculate this enhancement (see Su & Chesnavich DOI:10.1063/1.442828).

Figures: - figure 2: The figure and data will be much easier to comprehend if the dotted
lines are replaced by symbols (e.g. square, disk, triangle up/down) at the top of the corresponding peaks. currently the colours are too similar.

- figure 3: You should provide the geometry of the clusters as text files (pdb or xyz). Allows readers to visualise the results in a molecule viewer.

- figure 4: If possible combine the curves in figure 4 for each row, using different colours and symbols.

---

## Short Comment (SC2) · 18 Sep 2017

Author's response to the discussion paper:

**Electron-induced chemistry in microhydrated sulfuric acid clusters**

Jozef Lengyel[1,2], Andriy Pysanenko[1], Michal Fárník[1]

[1]J. Heyrovský Institute of Physical Chemistry v.v.i., Czech Academy of Sciences, Dolejškova 3, 18223 Prague, Czech Republic

[2]Institut für Ionenphysik und Angewandte Physik, Universität Innsbruck, Technikerstraße 25, 6020 Innsbruck, Austria

*Correspondence to*: Jozef Lengyel (jozef.lengyel@jh-inst.cas.cz), Michal Fárník (michal.farnik@jh-inst.cas.cz)

The manuscript describes mass spectrometric measurements and quantum chemical calculations on the sulphuric acid water system. The authors propose a caging mechanism in which an intermediate [(H2O)nH.HSO4-] species forms, before the neutral hydrogen atom is released. This mechanism applies probably only in cases where H2SO4 is charged directly by an e-. However, to my knowledge thermal electrons will most likely attach first to more abundant species. H2SO4 would then react with those ions and form HSO4-. If the authors have any references to more recent work that indicate direct charging of H2SO4 by free electrons in the Earth atmosphere they should cite them. H2SO4 is an important compound for the formation of aerosol particles, even if more recent research draws a more complex picture. This aspect of the manuscript fits therefore well into the scope of ACP. In general I agree with Ref 1 remarks about the quantum chemical calculations and this part should be extended.

We would like to thank the referee for reading our manuscript, his valuable comments and overall positive evaluation of our manuscript. Before going to his particular questions, we would like to provide some general comments. First, some of the reviewer's questions and comments concern the theory. Indeed, we try to answer them and incorporate his suggestions in revised manuscript, nevertheless, we would like to stress the point which we have made in our reply to reviewer #1: the major contribution of our present paper is the experiment and the calculations were performed only to provide a support for the unambiguous experimental observations and conclusions. Please, see our reply to reviewer #1, page 1 and 2.

Second, as outlined below, many of the reviewer's questions (points 1-8) originate probably from a slight confusion of our molecular beam method with the condensationchamber type experiments generally used in the aerosol community. Therefore, we provide an extensive explanation of the questioned experimental points for the reviewer below. However, we have not substantially extended the experimental part of our manuscript in this respect, since most of our arguments can be found in basic textbooks on molecular beams and clusters, e.g., [Scoles, G.: Atomic and molecular beam methods, Oxford University Press, 1988; Haberland, H.: Clusters of atoms and molecules. Springer series in chemical physics, vol. 52, Springer-Verlag, 1994; Pauly, H.: Atom, molecule, and cluster beams I & II. Springer-Verlag, 2000]. The conditions specific to our apparatus are then either outlined in the method section or in the cited publications.

Concerning the reviewer's question: "…to my knowledge thermal electrons will most likely attach first to more abundant species. H2SO4 would then react with those ions and form HSO4-…". The reviewer is correct and this has been pointed out in our manuscript on page 2: "*The cosmic rays are the principal source of ionization and free electrons in the upper troposphere and lower stratosphere, where the electron collisions can influence the gas-phase chemistry. Typically, the electrons are effectively thermalized to low energies ($\gtrsim 1$ eV) via multiple inelastic collisions (Campbell and Brunger, 2016) . At these low energies, they are rapidly captured by abundant molecules, in particular $O_2$.*" However, we focus on the mixed $H_2SO_4/H_2O$ clusters and use the electron attachment to ionize these species for the mass spectrometry which delivers some information also about the neutral clusters, their properties and elementary processes in them such as the acidic ionization. In addition, our experiments provide some information about the free electron attachment to these clusters. The main purpose of our study is the molecular level insight into these species and processes, which can be further relevant for atmospheric chemistry.

**1) To what temperature does the He/H2SO4/H2O gas mixture drop after nozzle?**

It ought to be noted that a supersonic expansion is generally a non-equilibrium process for which the concept of temperature is not well defined. Due to the gas rarefaction in the expansion, the collisions cease at some point (freezing/quitting surface) and the molecules and clusters than fly in the vacuum without further interactions. Typically the molecules undergo $\sim 10^4$ collisions in the expansion and they all happen within ~20 nozzle radii [Pauly, H.: Atom, molecule, and cluster beams I & II. Springer-Verlag, 2000]. In our case this represents ~2 mm from the nozzle throat, and the molecules pass through this region in a few microseconds. Please, see also our answer to referee #1, point I). A temperature can still be used in supersonic expansion to describe the species in the beam, however, one has to be aware that

different degrees of freedom require different number of collisions to cool and equilibrate. Therefore we can speak about rotational, vibrational and translational temperatures which can all be quite different (sometimes even the concept of parallel and perpendicular translational temperature is introduced despite the temperature not being a vector).

Determination of the cluster temperature experimentally is very challenging and has be accomplished only for a few special cases, e.g., for rare gas clusters (~35 K for $Ar_N$) and a few other species by electron diffraction [Farges, J., et al.: Structure and temperature of rare gas clusters in a supersonic expansion. Surf. Sci., 106, 95–100, 1981; Farges, J., et al.: Noncrystalline structure of argon clusters. II. Multilayer icosahedral structure of $Ar_N$ clusters $50<N<750$. J. Chem. Phys., 84, 3491–3501, 1986]. For the special case of superfluid helium nanodroplets the vibrational-rotational temperature of a molecule deposited in the cluster was measured spectroscopically giving the temperature of 0.37 K [Hartman, M., et al.: Rotationally resolved spectroscopy of $SF_6$ in liquid helium clusters: A molecular probe of cluster temperature. Phys. Rev. Lett., 75, 1566–1569, 1995]. For small clusters spectroscopic methods can be used in some cases. Nevertheless, for most of the larger molecular clusters, we have to rely on model approaches combined with an indirect experimental evidence to estimate their temperatures, e.g., the evaporative ensemble theory [Klots, C. E.: Evaporative cooling. J. Chem. Phys., 83, 5854–5860, 1985; Klots, C. E.: Temperatures of evaporating cluster. Nature, 327, 222–223, 1987; Klots, C. E.: Kinetic methods for quantifying magic. Z. Phys. D, 21, 335–342, 1991]. A relaxation model was introduced more recently to estimate the temperatures of water clusters [Brudermann, J., et al.: Isomerization and melting-like transition of size-selected water nonamers. J. Phys. Chem. A, 106, 453–457, 2002] and methanol clusters [Steinbach, C., et al.: Isomeric transitions in size-selected methanol hexamers probed by OH-stretch spectroscopy. Phys. Chem. Chem. Phys., 8, 2752–2758, 2006]. Further non-equilibrium numerical models were employed to analyze the cluster temperatures in seeded supersonic expansions of water vapor with Ar and Ne [Jansen, R., et al.: Nonequilibrium numerical model of homogeneous condensation in argon and water vapor expansions. J. Chem. Phys., 132, 244105, 2010; Gimelshein, N., et al.: The temperature and size distribution of large water clusters from a non-equilibrium model. J. Chem. Phys., 142, 244305, 2015], which were also studied spectroscopically by sodium doping method [Buck, U., et al.: A size resolved investigation of large water clusters. Phys. Chem. Chem. Phys., 16, 6859–6871, 2014; Zeuch, T., Buck, U.: Sodium doped hydrogen bonded clusters: Solvated electrons and size selection. Chem. Phys. Lett., 579, 1–10, 2013]. Different models revealed

temperatures between 70 K and 200 K for the water clusters, depending on cluster size and expansion conditions.

It ought to be mentioned that there are special Laval-type nozzle expansions where the equilibrium conditions can be reached, and experiments were designed to study nucleation in the nozzle and post-nozzle flows [Wyslouzil, B. E., et al.: Binary condensation in a supersonic nozzle. J. Chem. Phys., 113, 7317–7329, 2000; Manka, A., et al.: Freezing water in no-man's land. Phys. Chem. Chem. Phys., 14, 4505–4516, 2012; Wyslouzil B. E, Wölk, J.: Overview: Homogeneous nucleation from the vapor phase–The experimental science. J. Chem. Phys. 145, 211702, 2016; Schläppi, B., et al: A pulsed uniform Laval expansion coupled with single photon ionization and mass spectrometric detection for the study of large molecular aggregates. Phys. Chem. Chem. Phys., 17, 25761–25771, 2015; Ferreiro, J. J., et al.: Observation of propane cluster size distributions during nucleation and growth in a Laval expansion. J. Chem. Phys., 145, 211907, 2016; Chakrabarty, S., et al.: Toluene cluster formation in Laval expansions: Nucleation and growth. J. Phys. Chem. A, 121, 3991–4001, 2017]. However, the molecular beam experiments are not well suited to investigate the nucleation of aerosols in general. In molecular beam experiments, the expansion conditions are tuned to produce the desired species and then other experiments are performed with them. We did not attempt to study the nucleation of sulfuric acid but our aim was rather to produce the hydrated sulfuric acid clusters and investigate them by electron attachment and mass spectrometry.

It should be mentioned that the molecular beam experiments can also yield some information useful for molecular level understanding of the nucleation –e.g., in our previous experiments the cross sections of water clusters for pickup of various atmospheric molecules from gas phase could be measured which is essential for the cluster growth [Lengyel, J., et al.: Uptake of atmospheric molecules by ice nanoparticles: Pickup cross sections. J. Chem. Phys., 137, 034304, 2012]. However, this is not the subject of our present publication.

We provide this extensive answer and literature review, to illustrate that the question about the temperature in the expansion is not a trivial one and cannot be easily answered –in fact, this question has been answered so far only for a few specific systems. For the present system, the cluster temperature has not been measured experimentally and the above mentioned theoretical approaches are far beyond the scope of our present research. For pure water clusters the temperature would probably be around 100 K for our present expansion conditions. However, it is well known that already a small admixture of other gas in the expansion can change the conditions significantly, and consequently the resulting

temperature. Therefore we would like to refrain from any speculations about the cluster temperature at this point.

2) What relative humidity would be reached in the cluster formation zone after the nozzle?

Under our most humid conditions (figure 1c in the main paper) the partial water vapor pressure was approximately 42 mbar while the partial vapor pressure of $H_2SO_4$ was 2.04 mbar and we used 2 bar of He buffer gas. Please, see also our reply to Reviewer #1, page 3, last paragraph. RH corresponds to the ratio of the partial pressure of water vapor to the equilibrium vapor pressure of water at the temperature in the expansion. The clusters are formed still in the nozzle within few millimeters from the nozzle throat. The reservoir and nozzle temperatures were 453 K and 458 K, however, these temperatures are irrelevant for the actual temperatures in the gas flow which cools down by the supersonic expansion and the temperature can be as low as 100 K or even lower, leading to supersaturation in the expansion which in turn leads to the cluster formation [Pauly, H.: Atom, molecule, and cluster beams I & II. Springer-Verlag, 2000]. The temperature is not known for the present system, as discussed above, therefore we cannot give any RH value. The partial pressures of component molecules and buffer gas are sufficient to determine the cluster sizes and compositions in supersonic expansions (together with other expansion conditions such as the nozzle temperature, size and shape).

3) How long is the reaction time to form these clusters (before ionisation)?

There are several time-windows in our experiment. First, there is the time during which the collisions occur and the clusters are generated. This has been explained above that the collisions occur within ~2 mm from the nozzle throat, and the molecules pass through this region in about 2 μs (based on the cluster velocity ~1.0-1.5×10$^3$ ms$^{-1}$ which can be actually measured on our apparatus using a pseudo-random chopper and cross-correlation method [Fedor, J., et al.: Cluster cross sections from pickup measurements: Are the established methods consistent? J. Chem. Phys., 135, 104305, 2011; Lengyel, J., et al.: Uptake of atmospheric molecules by ice nanoparticles: Pickup cross sections. J. Chem. Phys., 137, 034304, 2012]). Thus the first "reaction time" in which the clusters are generated is ~2 μs.

Then the collisions cease completely. After the skimmer (~1.5 cm from the nozzle) the beam enters high and ultra-high vacuum regions (10$^{-6}$-10$^{-10}$ mbar) where the mean free path for molecules exceeds several meters, i.e., the clusters fly undisturbed, isolated, not interacting with anything (except for the black-body-radiation photons from the apparatus walls which are at room temperature). The distance to the ionization volume of the TOF

spectrometer is ~1.5 m which the clusters fly in about 1.5 ms. During this time only evaporation of molecules from the cluster can occur (if the clusters were generated in excited states) or spontaneous intra-cluster reactions can happen, e.g., the acidic dissociation. However, we believe that this process occurs already during the cluster generation, as soon as the sulfuric acid molecule is surrounded by enough water molecules in the cluster. So the second "reaction time" of 1.5 ms is rather an idle time. Then the ionization by electron beam occurs.

**4) How long is the reaction time the charged clusters (after ionisation)?**

The electron beam is pulsed for 2 μs during which the clusters can interact with the electrons. After the electron pulse there is a delay of 0.5 μs before the extraction pulse for ions is switched on. During this delay all remaining free electrons leave the ionization volume so that the spectra cannot be disturbed by any effects caused by these electrons accelerated by the extraction voltage. Then the ions are accelerated by an 8 kV pulse into the TOF where the total flight path to the detector is 95 cm. The flight time between the cluster ionization and detection represents several tens of microseconds –the exact value depends on applied voltages, and the cluster mass to charge ratio. The pressure in the TOF is below $10^{-8}$ mbar, i.e., the ions do not collide with any residual gas molecules. Also the cluster density in the beam is so low that any collisions of the nascent cluster ions with other clusters or molecules in the beam are highly unlikely.

In summary, the "reaction time" after the cluster ionization is several tens of microseconds (depending on the mass to charge ratio). However, during this time the cluster ions do not collide with any other species and thus only intracluster reactions can occur. If the cluster ion fragmentation would occur during the time after the cluster extraction and before it enters the reflectron mirror, it would be reflected in the position and shape of the corresponding mass peaks [Wei, S., Castleman, A. W.: Using reflectron time-of-flight mass spectrometer techniques to investigate cluster dynamics and bonding. Int. J. Mass. Spectrom. Ion Processes, 131, 233–264, 1994]. We did not observe any of these effects in the mass spectra, therefore we excluded metastable fragmentation of the clusters after the ionization. Indeed, we observed the cluster ion fragments in the mass spectra, but the fragmentation had to be fast, completed between the ionization and extraction pulse, i.e. within 0.5 μs (taking into account also the ionization pulse duration, the maximum time available for this fragmentation could be 2.5 μs).

**5) Is it possible that some water is condensing on the cluster after the ionisation?**

The cluster formation, which includes also the addition of water molecule, is strictly connected only to the supersonic expansion. As described above, the TOF chamber is ~1.5 m downstream from the nozzle chamber divided by several differentially pumped vacuum chambers and there are no further collisions possible in the ultrahigh vacuum in our TOF chamber (the corresponding pressure of ~$10^{-8}$ mbar corresponds to the mean free path in the order of 10 km for molecules).

**6) Could you determine the lifetime of the intermediate by adjusting the reaction time?**

We are not sure what the Reviewer means by the "intermediate" and "reaction time" but we have probably answered this question above, point 4). If some intermediate is generated after the electron attachment which then fragments to the final cluster ion, the fragmentation has to occur within the 0.5 μs delay between the electron pulse and the ion extraction (at most 2.5 μs if we take into account also the ionization pulse duration). Any metastable decay which would occur after the ion extraction during the flight time to the reflectron mirror would be recognized in the spectral peak position and shape. Since no such effects were observed in the spectra the intermediate lifetime must be shorter than 0.5 μs. We could make this delay time even shorter, however, some unwanted effects from residual free electrons could disturb the spectrum. Therefore this is the shortest time window we could provide at the moment.

**7) I assume the charged clusters are accelerated by an electric field in a low pressure environment, is evaporation and fragmentation a potential problem in your setup?**

We believe that we have answered this question already above. If the cluster mass changes after its extraction during its flight time to the reflectron mirror, the fingerprint of the mass change can be unambiguously recognized in the spectrum [Wei, S., Castleman, A. W.: Using reflectron time-of-flight mass spectrometer techniques to investigate cluster dynamics and bonding. Int. J. Mass. Spectrom. Ion Processes, 131, 233–264, 1994]. There was no evidence for such metastable fragmentation in our spectra.

**8) How do your dissociation constants compare to bulk phase sulphuric acid?**

The dissociation constant is a characteristics of a macroscopic bulk systems in equilibrium. For individual isolated clusters in vacuum we cannot measure the dissociation constants in our experiment. From our mass spectra we can make some conclusions about the acidic dissociation of sulfuric acid in water clusters. In principle we can answer the question: how many water molecules are needed to acidically dissociate a sulfuric acid molecule? Actually,

this question has been treated by similar methods in the past for nitric acid [Kay, B. D., et al.: Studies of gas-phase clusters: The solvation of $HNO_3$ in microscopic aqueous clusters, Chem. Phys. Lett., 80, 469, 1981] (see also our answer to point 3) of reviewer #1. Such question cannot be solved in the bulk where the dissociation constant characterizes the macroscopic system. However, it is a fundamental question and the answer to this question provides molecular level understanding to the processes occurring in the bulk.

9) Data Statement is missing. https://www.atmospheric-chemistry-and-physics.net/about/data_policy.html

The data statement has been added at the end of the manuscript.

10) A brief overview of current developments in the quantum chemical treatment of H2SO4-H2O clusters is missing.

As mentioned above, the major contribution of our present paper is the unique experiment. The calculations were performed to provide a support for the experimental conclusions. The major conclusions, e.g., about the acidic dissociation or fragment caging in the clusters could be derived essentially just based on the experimental evidence. Please, see our reply to the reviewer #1, page 1-2 for more details. Besides, we would like to point out that our benchmark calculations proved that the used computational approach is in reasonable agreement with the higher-level *ab initio* methods. These benchmark calculations are now presented in supporting information and also discussed in our reply to the reviewer #1 on p. 2.

Since the present manuscript is not a theoretical paper, and as experimentalists we rather focus on the previous experimental evidence directly relevant to the investigated phenomena, we believe that an extensive overview of the previous quantum chemical calculations of the $H_2SO_4$-$H_2O$ system would unnecessarily extend the theory-related part of our manuscript and distract the reader from the main message. We believed that we have covered most of the recent theoretical papers directly relevant to our investigations in our references. However, we could miss some important contributions and we would be happy to add some more references, if the reviewer provides some explicit suggestions what we missed in our survey.

11) P 1 L 19: I think aerosols should be replaced with clusters as the paper shows only results for clusters containing a few sulphuric acid molecules

We followed the reviewer's suggestion and replace "aerosols" with "clusters".

12) P 1 L 27: I suggest to replace neutral structure by undissociated structure and (optionally) ion pair structure by dissociated structure (ps: probably solved by following referee 1 request)

We followed the suggestion of both referees and the respective terminology has been changed to avoid any confusion.

13) P2 L 3-9: Is the Sulphuric acid charging by e- capture actually important for the production of charged sulphuric acid cluster (especially in the context of boundary layer nucleation events)? Some older and more recent literature is missing in the context of ion induced nucleation Raes and Jansen 1985 DOI: 10.1016/0021-8502(85)90028-X (or even earlier), a more recent one would be Kirkby et al 2011 (DOI: 10.1038/nature10343) or the most recent for H2SO4 water Kürten et al 2016 (10.1002/2015JD023908). There are also field observations that try to estimate the fraction of ion induced nucleation in the total nucleation rate (Hirsiko et al 2011, doi:10.5194/acp-11-767-2011).

As already mentioned above, we do not focus on the nucleation of atmospheric sulfuric acid aerosols. We investigate the properties of the individual mixed $H_2SO_4$-$H_2O$ clusters which have not been observed by mass spectrometric experiments before. Therefore we have avoided an extensive discussion of the atmospheric nucleation of sulfuric acid aerosols in the introduction, since we found it unnecessarily lengthening our paper and distracting the reader from the main topic. Nevertheless, we have now added a brief overview of the nucleation in the introduction on page 2 citing the above publications:

14) P2 L14-15: Charged sulphuric acid clusters in the atmosphere where already measured before, example literature Eisele, 1989 DOI: 10.1029/JD094iD02p02183 I have the impression this sentence is out of context and should be moved to the previous paragraph.

The introduction has been changed as outlined above.

15) Methods: P4 L 2: What was the criterion for choosing the clusters for reoptimisation? How many isomers were used?

We used previously observed energetic minima from literature as the initial structures and equilibrated them in molecular dynamics runs. From the MD simulation, several structures were randomly taken and re-optimized at the M06-2X/aug-cc-pVDZ level. Altogether 8 different isomers on average were optimized from various starting structures for each cluster type $((H_2SO_4)_{1-2}(H_2O)_{0-5}$, $(H_2SO_4)_{0-1}(H_2O)_{0-5}HSO_4^-)$ including the hydrogen-bonded, $H_2SO_4\cdots H_2O$, and ion-pair, $HSO_4^-\cdots H_3O^+$, structures in neutral clusters. Only the most stable

isomers were considered for further calculations. The section '*2 Experimental and theoretical methods*' has been extended with the aforementioned discussion.

16) Results: P4 L 17: The results of the quantum chemical calculations should be compared to previous work.

The comparison with the previous theoretical work has been added to the discussion concerning the binary nucleation of $H_2SO_4$/$H_2O$ clusters. As an example, the calculated free energies of the addition of water molecule were in reasonable agreement with the literature values [Kurtén, T., et al.: Quantum chemical studies of hydrate formation of $H_2SO_4$ and $HSO_4^-$. Boreal Environ. Res., 12, 431, 2007; Loukonen, V., et al.: Enhancing effect of dimethylamine in sulfuric acid nucleation. Atmos. Chem. Phys., 10, 4961, 2010; Henschel, H., et al.: Hydration of atmospherically relevant molecular clusters: Computational chemistry and classical thermodynamics. J. Phys. Chem. A, 118, 2599, 2014]. The seemingly different values in free energies were caused by different definitions of the nucleation event in the calculated chemical reactions:

$$H_2SO_4(H_2O)_{n-1} + H_2O \rightarrow H_2SO_4(H_2O)_n \qquad \Delta_r G^0(1)$$

$$H_2SO_4 + nH_2O \rightarrow H_2SO_4(H_2O)_n \qquad \Delta_r G^0(2)$$

In the present paper, we calculate the free energy $\Delta_r G^0(1)$ for an addition of single water molecule to a preexisting cluster. The free energy can be calculated also from the accumulation of all molecular component to the cluster $\Delta_r G^0(2)$. The former way of calculation, $\Delta_r G^0(1)$, is more illustrative for our discussion. When we calculated $\Delta_r G^0(2)$, it was in a good agreement with the previously published data as shown in Table 1.

**Table 1**: Free energies (in kcal mol$^{-1}$, at $T$=298K and $p^0$=1atm) of binary nucleation of $H_2O$/$H_2SO_4$ clusters

| $n =$ | $\Delta_r G^0(1)$ our results | $\Delta_r G^0(1)$ Kurtén, et al. | $\Delta_r G^0(2)$ our results | $\Delta_r G^0(2)$ Henschel, et al. | $\Delta_r G^0(2)$ Loukonen, et al. |
|---|---|---|---|---|---|
| 1 | −2.7 | −2.81 | −2.7 | −2.60 | −2.93 |
| 2 | −1.7 | −1.87 | −4.4 | −4.40 | −6.26 |
| 3 | −0.8 | −2.37 | −5.2 | −5.83 | −7.11 |
| 4 | −2.3 | −0.90 | −7.5 | −7.05 | −8.11 |
| 5 | −1.2 | − | −8.7 | −6.81 | −10.01 |

The corresponding discussion has been added to the manuscript, and the respective chemical equations included in a new Table 1 in the manuscript.

17) P8 L 21: could this be quantified more? Is the cross section and e- concentration high enough so that electron attachment in the troposphere is a major source of HSO4-? What is the free e- concentration in the troposphere?

The cosmic ray ionization rate varies between about 2 ion pairs $cm^{-3}$ $s^{-1}$ close to Earth's surface and 40 ion pairs $cm^{-3}$ $s^{-1}$ at the top of the troposphere [Carslaw, K. S., et al.: Cosmic rays, clouds, and climate, Science, 298, 1732–1737, 2002]. The observed density of free electrons is very low in stratosphere around $5\times10^3$ $cm^{-3}$ [Smith, D., Adams, N. G.: Elementary plasma reactions of environmental interests. Top. Curr. Chem. 89, 1–43, 1980] because they rapidly interact with abundant molecules, in particular $O_2$. For the lower altitudes, the number of the free electrons is significantly reduced. Therefore we expect a minor importance of free electrons for tropospheric ion chemistry. The tropospheric $HSO_4^-$ ions are most likely formed in reaction of gas-phase sulfuric acid and various molecular anions, such as $O_2^-$. At higher altitudes the contribution of free electrons is larger, therefore we have suggested that the free electron attachment may actually contribute. However, our major focus is the molecular level understanding to the mixed $H_2SO_4/H_2O$ clusters and the electron attachment process to them. We are not qualified to make speculations about the actual contribution of such processes, however, the corresponding sentence can stimulate the interest of scientists modelling the atmospheric processes.

18) P9 L 3: How was that calculated? You probably used all structures for a given isomer, please be a bit more precise here.

The computational procedure was described in detail in the section '*2 Experimental and theoretical methods*', page 4, line 3 (in the original manuscript): "Only the most stable isomers were considered for further calculations."

19) Conclusion:  P10 L 16-17: The dipole in the dissociated cluster could also enhance the uptake of ions such as NO3-. With the quantum chemical calculations you could calculate this enhancement (see Su & Chesnavich DOI:10.1063/1.442828).

The reviewer is correct and the dipole or positive charge in the dissociated cluster could also enhance the uptake of ions such as $NO_3^-$. This could be calculated theoretically. However, we would like to stress again, that the main value of our contribution is the solid experimental evidence from an advanced experiment with particles in the molecular beam. To perform such

experiment for the proposed $NO_3^-$ ion attachment would require substantial changes in our experimental arrangement and would be difficult –for example, already the initial question of a good source for an intense $NO_3^-$ ion beam would be difficult to solve. This goes far beyond not even our present publication but also our near future experimental plans. But we thank the reviewer for an interesting input for some future plans.

Our present study is focused on the binary $H_2SO_4/H_2O$ clusters. Apparently the detection of these hydrated clusters seems to be not trivial for laboratory experiments since they have not been detected in the aerosol chamber experiments so far [Kirkby, J., et al.: Role of sulphuric acid, ammonia and galactic cosmic rays in atmospheric aerosol nucleation, Nature, 476, 429, 2011, Kürten, A., et al.: Neutral molecular cluster formation of sulfuric acid-dimethylamine observed in real time under atmospheric conditions. Proc. Natl. Acad. Sci. USA, 111, 15019, 2014]. Here we have demonstrated an efficient way of generating these hydrated sulfuric acid clusters using the molecular beam technique, and characterized them by the electron attachment mass spectrometry. Adding additional calculations of ternary complexes with $NO_3^-$ ion to our present manuscript would be confusing without any experiment connected to this topic.

20) figure 2: The figure and data will be much easier to comprehend if the dotted lines are replaced by symbols (e.g. square, disk, triangle up/down) at the top of the corresponding peaks. currently the colours are too similar.

The symbols have been added to Figure 2.

21) figure 3: You should provide the geometry of the clusters as text files (pdb or xyz). Allows readers to visualise the results in a molecule viewer.

The respective geometries have been added to the Supplement.

22) figure 4: If possible combine the curves in figure 4 for each row, using different colours and symbols

We agree with the reviewer that there seems not to be much information in figure 4, except that the energy dependent ion yields look essentially all the same. However, that is exactly our message. As outlined also in the text, the ion yield curves were identical for all the mass peaks (within the experimental errors). Figure 4 should illustrate this point for a few examples (there are plenty of electron energy dependencies measured for other mass peaks, and we have added some more examples in the supporting information for illustration). If we plotted all the energy dependencies in one single plot (normalized) they would be just overlapping. We

believe that showing the individual plots separately provides a clearer picture, and the reader can also see the different signal intensities and the level of the noise in the data. This presentation of figure 4 provides a clear picture of what is meant in the text by "the same energy dependencies". We believe that this message would not be so clear in the representation suggested by the reviewer, therefore we would like to keep the present form of figure 4.

---

## Author Comment (AC1) · 6 Oct 2017

The comment was uploaded in the form of a supplement:
https://www.atmos-chem-phys-discuss.net/acp-2017-573/acp-2017-573-AC1-supplement.pdf

---

## Author Response (AR1)

Author's response to the discussion paper:

**Electron-induced chemistry in microhydrated sulfuric acid clusters**

Jozef Lengyel[1,2], Andriy Pysanenko[1], Michal Fárník[1]

[1]*J. Heyrovský Institute of Physical Chemistry v.v.i., Czech Academy of Sciences, Dolejškova 3, 18223 Prague, Czech Republic*
[2]*Institut für Ionenphysik und Angewandte Physik, Universität Innsbruck, Technikerstraße 25, 6020 5 Innsbruck, Austria*

*Correspondence to: Jozef Lengyel (jozef.lengyel@jh-inst.cas.cz), Michal Fárník (michal.farnik@jh-inst.cas.cz)*

**Reply to the interactive comment of Referee #1:**

We would like to thank the referee for his valuable comments and overall positive evaluation of our manuscript. Before addressing his particular points in more detail, we would like to make a general comment.

We would like to stress that the major contribution of our present paper is the experiment. The calculations were performed to provide a support for the experimental conclusions. The major conclusions, e.g., about the acidic dissociation or fragment caging in the clusters could be derived essentially just based on the experimental evidence. Actually, such conclusions were derived previously for similar systems (nitric acid–water clusters) just from the experimental mass spectra in the early work of Castleman's group [Kay, B. D., Hermann, V., and Castleman Jr., A. W.: Studies of gas-phase clusters: The solvation of $HNO_3$ in microscopic aqueous clusters, Chem. Phys. Lett., 80, 469, 1981]: the number of water molecules needed to acidically dissociate an $HNO_3$ molecule in $HNO_3(H_2O)_N$ clusters was derived from the shape of the $HNO_3(H_2O)_nH^+$ mass spectra (see also our work: [Lengyel, J., Pysanenko, A., Kočišek, J., Poterya, V., Pradzynski, C. C., Zeuch, T., Slavíček, P., and Fárník, M.: Nucleation of mixed nitric acid-water ice nanoparticles in molecular beams that starts with a $HNO_3$ molecule, J. Phys. Chem. Lett., 3, 3096, 2012]). Recently, we have shown that for the $(HNO_3)_M(H_2O)_N$ clusters the conclusions drawn from the mass spectra are actually in excellent agreement with the theoretical calculations [Lengyel, J., Ončák, M., Fedor, J., Kočišek, J., Pysanenko, A., Beyer, M. K., and Fárník, M.: Electron-triggered chemistry in $HNO_3/H_2O$ complexes, Phys. Chem. Chem. Phys., 19, 11753, 2017]. Therefore, in the present

case we implement the theoretical calculations to support our conclusions drawn from the experimental evidence –and once again the experiment and theory are in excellent agreement.

We understand that our present level of theory might not exceed the theory level required for a stand-alone theoretical paper. However, that was not our ambition –we performed the calculations at the level accessible to our experimental group and they were in agreement with the experiment. Therefore we published them alongside with the experiment as they can provide more (pictorial) insight into what is actually happening in the clusters. It ought to be mentioned that even the calculations performed at the highest possible level of theory are not guaranteed to deliver a reliable picture of what is going on in the real system, unless they are backed up by some experimental evidence.

Besides, we would like to stress that our benchmark calculations proved that the used computational approach using double zeta basis set are in reasonable agreement with the higher-level *ab initio* methods. **Table 1** summarizes the benchmark calculations of electron affinity of $HSO_4$, ionization potential of $H_2SO_4$, and reaction enthalpies for deprotonation of gas-phase $H_2SO_4$ calculated at different levels of theory. The M06-2X/aug-cc-pVDZ energies are comparable with the CCSD/aug-cc-pVDZ values with the exception of the $IP(H_2SO_4)$. The comparison of double-zeta with triple-zeta basis sets of the M06-2X functional shows that there is essentially constant shift from the experimental values and therefore we do not expect any significant shift in reaction energies even upon hydration. The calculated reaction enthalpies for deprotonation of gas-phase $H_2SO_4$ are in good agreement with the experimental value. The error of the DFT method is 0.1-0.2 eV. Please note that, in the present work, chemical trends with respect to hydration are of the main concern, and a possible systematic shift of few tenths of eV does not influence our conclusions.

**Table 1:** Electron affinity of $HSO_4$, ionization potential of $H_2SO_4$, and enthalpy of deprotonation at various levels of theory (all in eV). DZ and TZ represent aug-cc-pVDZ and aug-cc-pVTZ, respectively. Enthalpies were calculated at 298.15 K within the harmonic approximation.

|  | B3LYP/DZ | M06-2X/DZ | M06-2X/TZ | MP2/DZ | CCSD/DZ | Experiment |
|---|---|---|---|---|---|---|
| $EA(HSO_4)$ | 4.69 | 4.92 | 5.01 | 5.21 | 4.92 | $4.75\pm0.10^a$ |
| $IP(H_2SO_4)$ | 11.4 | 11.6 | 11.8 | 12.4 | 12.5 | $12.4\pm0.05^b$ |
| $\Delta H(H_2SO_4{\rightarrow}H^++HSO_4^-)$ | 13.7 | 13.5 | 13.5 | 13.4 | 13.6 | $13.4\pm0.24^a$ |

[a] Wang, X.B., Nicholas, J.B., Wang, L.S.: Photoelectron spectroscopy and theoretical calculations of $SO_4^-$ and $HSO_4^-$: Confirmation of high electron affinities of $SO_4$ and $HSO_4$, J. Phys. Chem. A, 104, 504, 2000.

[b] Snow, K.B., Thomas, T.F.: Mass spectrum, ionization potential, and appearance potentials for fragment ions of sulfuric acid vapor, Int. J. Mass Spectrom. Ion Processes, 96, 49, 1990.

Now we would like to address the actual referee's points:

I) My main question is, why does the concentration of clusters with 3 or more H2SO4 molecules drop when more water is added (e.g. figure 1)? According to both classical thermodynamics and quantum chemical calculations (as well as chemical common sense), water should promote clustering of sulfuric acid significantly. Thus one would assume that the concentration of larger clusters would increase when the water content goes up. Or to put it another way, typically the nucleation rate increases with increasing RH for constant [H2SO4] and T. Does perhaps the absolute H2SO4 concentration (which the authors don't actually report) decrease from the low-water runs to the high-water runs?

The referee is, indeed, correct that increasing water concentrations promotes the sulfuric acid clustering in condensation chamber type experiments where equilibrium conditions can be reached. However, such conditions are far from our experimental method of the cluster generation. In supersonic expansions the clusters are generated in a very limited space and time span. Typically the molecules undergo ~$10^4$ collisions in the expansion and they all happen within ~20 nozzle radii. In our case this represents ~2 mm from the nozzle throat, and the molecules pass through this region in a few microseconds. After that (freezing/quitting surface) the molecules undergo no more collisions and the clusters which have been formed essentially do not change until the interaction with the electron beam in the mass spectrometer 2.5 m downstream from the nozzle. An important aspect is also the strong cluster cooling in the expansion due to the inelastic collisions with the buffer gas –the buffer gas atoms carry away the energy in their kinetic energies transforming the internal cluster energy into the kinetic energy of the gas flow in the direction of the beam. The clusters can be cooled by this mechanism to very low temperatures (e.g. for pure water clusters temperatures below 100 K can be routinely achieved). Due to the gas rarefaction the supersonic expansion is a non-equilibrium process and the cluster composition is determined by the collisions between the constituent molecules in the expansion and by the cooling in the collisions with the buffer gas.

In the spectra in figure 1 in the main paper, the He buffer gas pressure is kept constant at ~2 bar and we control the $H_2O$: $H_2SO_4$ ratio in the vapor by controlling the temperature of the reservoir $T_R$ containing the sulfuric acid. Our aim was to obtain the mixed clusters. Panel (a) in figure 1 corresponds to concentrated sulfuric acid (98.5%) in the reservoir at $T_R = 453$ K. The partial vapor pressure of water and sulfuric acid under these conditions was 3.09 mbar and 2.12 mbar, respectively [Perry, R. H., Green D. W., Maloney, J. O.: Perry's chemical engineers' handbook, 7th, MacGraw-Hill, New York, 1997], corresponding to the mole

fractions indicated in the figure. These conditions yielded almost exclusively the pure $H_2SO_4$ clusters. The spectrum in panel (b) corresponded to the sulfuric acid concentration of 98.0% at the same $T_R$, i.e., the partial vapor pressures of $H_2O$ and $H_2SO_4$ were 4.92 mbar and 2.04 mbar, respectively [Perry, R. H., Green D. W., Maloney, J. O.: Perry's chemical engineers' handbook, 7th, MacGraw-Hill, New York, 1997]. In panel (c) we added more water directly into the carrier gas using the humidifier –this is a new very successful method introduced by our group just recently for microhydration of biomolecules [Kočišek, J., Pysanenko, A., Fárník, M., and Fedor, J.: Microhydration prevents fragmentation of uracil and thymine by low-energy electrons, J. Phys. Chem. Lett., 7, 3401, 2016]. This way we increased the partial water vapor pressure to approximately 42 mbar while the partial vapor pressures of $H_2SO_4$ remained 2.04 mbar. This finally yielded the substantial hydration and the mixed clusters.

Thus the water concentration increases from (a) to (c). First, there is very little water and there are mostly collisions between $H_2SO_4$ molecules and He generating the pure $(H_2SO_4)_N$ clusters. Increasing the water concentration, the collisions with water molecules become more frequent and some $H_2O$ molecules stick to the sulfuric acid and the mixed clusters appear. At the same time the collisions between $H_2SO_4$ molecules become less frequent resulting in smaller $(H_2SO_4)_N$ clusters.

Under equilibrium conditions in nucleation chambers, the most likely mechanism how the water promotes clustering is by generating mixed clusters in the first place, and subsequently the water is replaced with $H_2SO_4$ molecules in these clusters. Therefore, essentially only sulfuric acid clusters without water can be observed in the nucleation chamber type experiments. However, this is not the case in our supersonic expansions where the collisions cease after a short time before all water can be replaced in collisions with $H_2SO_4$ molecules. Our aim was to generate and investigate the elusive mixed clusters as the early stage in the sulfuric acid nucleation upon humid conditions. The molecular beams are an ideal tool for such type of experiment. The hypothesis of $H_2SO_4$ replacing water molecules could be also tested in our molecular beam experiment by pickup of $H_2SO_4$ on pure $(H_2O)_N$ clusters, however, it is technically very demanding far beyond our present experiment.

II) My main suggestion is that the authors add some calculations on the (H2O)nH2SO4-radical anionic clusters to support their extensive speculation on "H caging" and similar effects. While I understand their reluctance to work with larger open-shell clusters (with more than one H2SO4/HSO4- moiety), the (H2O)nH2SO4- system with small (e.g. 1-4) n is certainly treatable at the UM06-2X/aug-cc-pVDZ level, and while the resulting energies may not be as accurate as for the closed-shell systems, the structures would certainly be good enough to investigate the "caging" phenomenon the authors repeatedly speculate about.

We followed the suggestion of the referee and calculated the negatively charged $(H_2O)_nH_2SO_4^-$ clusters system with $n = 1-5$. **Fig. 1** represents the most stable energy isomers of $(H_2O)_nH_2SO_4$ ($n= 0-5$) clusters (figure 3 of the main paper) re-optimized as the negative ions at the M06-2X/aug-cc-pVDZ level of theory. Comparing the neutral and corresponding anionic structures, the main difference was reducing a dihedral angle between two OH groups from ~158° to ~75° of the sulfuric acid molecule, which resulted in changing of the water molecule orientation in the mixed clusters. The re-optimization of the s1w5-c structure was always followed by spontaneous $H_3O$ formation.

[Figure]

**Figure 1.** Re-optimized neutral most stable energy isomers of $H_2SO_4(H_2O)_n$ ($n= 0-5$) clusters with both covalent-bonded (c) and ion-pair (ip) structures as anions. The clusters were optimized at the M06-2X/aug-cc-pVDZ level of theory. The corresponding neutral structures are shown in figure 3 of the main paper.

Although, we have performed the calculations as suggested by the referee, we are not convinced that any further conclusions about the hydrogen caging can be made from these calculations. It ought to be mentioned that the caging, in the present case, is rather a solid experimental observation than a speculation. The electron attachment to a molecule is in principle a dissociative process (DEA) even if a stable anion exists for the molecule (even for zero kinetic energy electrons) [Fabrikant, I. I., Eden, S., Mason, N. J., and Fedor, J.: Recent progress in dissociative electron attachment, Adv. At. Mol. Opt. Phys., 66, 545, 2017]. For sulfuric acid the $H_2SO_4^-$ ion does not exist –neither experimentally nor theoretically and the hydration does not seem to stabilize the small $(H_2O)_n\ H_2SO_4^-$ species sufficiently. Yet, the observation of these ions in the mass spectra is an unambiguous experimental fact, therefore the $H_2SO_4 + e^- \rightarrow HSO_4^- + H$ dissociation after the DEA process has to be hindered and the hydrogen must be caged by the solvent in order to observe the $(H_2O)_n H_2SO_4^-$ series in the spectrum. We do not wish to speculate what kind of structural arrangement the $(H_2O)_n H_2SO_4^-$ clusters assume, since there are probably numerous possibilities (there will be also a distribution of the neutral starting $(H_2O)_n H_2SO_4$ cluster configurations of which figure 3 in the main paper represents only some examples to illustrate that there are neutral structures where the water molecules can hinder the free hydrogen dissociation).

**Minor comments:**

1) In the abstract, the authors state that "the (H2SO4)m(H2O)nHSO4- clusters are formed after the dissociative electron attachment to the clusters containing the (HSO4□° u°u°uH3O+) ion-pair structure". This is a reasonable conclusion to draw from their data, but their reasoning is based on somewhat indirect evidence - I would modify the sentence to account for this, e.g. by adding a word "likely", or starting the sentence with "Our results indicate that" or something similar.

We have changed the corresponding sentence according to referee's suggestion. It now starts with "Our results indicate that…"

2) In the introduction, the authors call the sulfuric acid - water clusters where the sulfuric acid remains undissociated "neutral". While this is not wrong, it can lead to confusion, as also the ion-pair clusters (HSO4-: : :H3O+) are "neutral" in the sense of having a overall electrical charge of zero. I would thus recommend the authors use some other term to denote the undissociated clusters. (Later on they themselves use the term "covalently-bonded", which is

one option; "hydrogen-bonded molecular cluster" would be even more accurate but somewhat lengthy.)

We have changed the term "neutral" to "covalently bonded $H_2SO_4$" wherever possible in the main article according to the referee's suggestion.

3) On page 7, the authors talk about the "presumably larger dipole moment" of the ion-pair structures. They do not need to presume anything about dipole moments, as their quantum chemical calculations contain the dipole moments of all their structures – they should instead report (in the supplement) the dipole moments of all their global minima, and for the "borderline" cases where the molecular cluster and ion pair structures are close in energy, perhaps report dipole moments for the best structures of both cases. These data could then be used to see whether the reasoning is indeed correct or not.

The argument that the dipole moment of the cluster increases upon the ion pair generation in the cluster was used for the explanation of the mixed $HNO_3(H_2O)_N$ cluster mass spectra [Kay, B. D., Hermann, V., and Castleman Jr., A. W.: Studies of gas-phase clusters: The solvation of $HNO_3$ in microscopic aqueous clusters, Chem. Phys. Lett., 80, 469, 1981]. Essentially the same argument was used in the interpretation of recent experiments where the mixed $HCl(H_2O)_N$ clusters were deflected in electric fields [Guggemos, N., Slavíček, P., and Kresin, V. V.: Electric dipole moments of nanosolvated acid molecules in water clusters, Phys. Rev. Lett., 114, 43401, 2015] (and also earlier for $HNO_3(H_2O)_N$ cluster [Moro, R., Heinrich, J., and Kresin, V. V.: Electric dipole moments of nitric acid-water complexes measured by cluster beam deflection, AIP Conf. Proc., 1197, 57, 2009]). Although, it ought to be mentioned that in the $HCl(H_2O)_N$ case the theoretical calculations showed that the change in the cluster dipole moment upon the acidic dissociation was relatively small and could be overlapped by dynamic effects. In our present investigation of the mixed $H_2SO_4(H_2O)_N$ clusters, the calculations show very broad range of dipole moments (e.g., from 0.3 D to 4.6 D for $N = 5$ clusters) which depend rather on the cluster structure than on the acidic dissociation (see **Fig. 2** –also added in SI now). Most likely, not only the energy minimum structure but many different cluster structures are generated in the supersonic expansion.

[Figure]

**Figure 2.** Selected local minima of neutral, $H_2SO_4\cdots H_2O$, (a-d) and ion-pair, $HSO_4^-\cdots H_3O^+$, (e-h) structures in $H_2SO_4(H_2O)_5$ clusters and the corresponding dipole moments.

However, the increase of the peak intensities in mass spectra as a function of the number of water molecules $n$ in the clusters is an unambiguous experimental observation. This increase has to be connected with a change in some physical properties of the clusters with $n$. The obvious property which changes with $n$ is the occurrence of the ion pair from a certain $n$ on –please, see [Kay, B. D., Hermann, V., and Castleman Jr., A. W.: Studies of gas-phase clusters: The solvation of $HNO_3$ in microscopic aqueous clusters, Chem. Phys. Lett., 80, 469, 1981] for more detailed argumentation. In the present case it can be the presence of the $H_3O^+$ in the cluster which leads to the more efficient electron attachment and generation of the negative $(H_2O)_n HSO_4^-$ clusters, rather than a larger dipole moment of the zwitterionic clusters (which does not have to be larger as the present calculations suggest).

Thus, thanks to this referee's comment, we have actually discovered an interesting issue, which might have been not quite correct in the past and recent literature and might require some attention in the future. Therefore we have modified our arguments correspondingly (two sentences: page 7 line 10; page 10 line 17).

4) On page 9, the authors mention the "polarization of the second H2SO4 molecule when the HSO4- ion is generated" as the reason for the very exothermal formation of HSO4- H2SO4 (and neutral free H) from (H2SO4)(H2SO4)-. This is not in itself wrong, but a more

informative/illustrative way to phrase it would be that the HSO4-(H2SO4) cluster has a much higher binding energy (by tens of kcal/mol) than the (H2SO4)(H2SO4) cluster.

We have changed the corresponding sentence according to referee's suggestion.

5) Line 2 of section 2: "home-build" should be "home-built".

The phrase has been corrected.

6) Figure caption of figure 1: "decreasing H2O mole fraction" should presumably be increasing (as we go from a to b to c, x(H2O) goes up)

The referee is correct, we apologize for this confusion and change the corresponding text.

Author's response to the discussion paper:

**Electron-induced chemistry in microhydrated sulfuric acid clusters**

Jozef Lengyel[1,2], Andriy Pysanenko[1], Michal Fárník[1]

[1]J. Heyrovský Institute of Physical Chemistry v.v.i., Czech Academy of Sciences, Dolejškova 3, 18223 Prague, Czech Republic

[2]Institut für Ionenphysik und Angewandte Physik, Universität Innsbruck, Technikerstraße 25, 6020 Innsbruck, Austria

*Correspondence to*: Jozef Lengyel (jozef.lengyel@jh-inst.cas.cz), Michal Fárník (michal.farnik@jh-inst.cas.cz)

The manuscript describes mass spectrometric measurements and quantum chemical calculations on the sulphuric acid water system. The authors propose a caging mechanism in which an intermediate [(H2O)nH.HSO4-] species forms, before the neutral hydrogen atom is released. This mechanism applies probably only in cases where H2SO4 is charged directly by an e-. However, to my knowledge thermal electrons will most likely attach first to more abundant species. H2SO4 would then react with those ions and form HSO4-. If the authors have any references to more recent work that indicate direct charging of H2SO4 by free electrons in the Earth atmosphere they should cite them. H2SO4 is an important compound for the formation of aerosol particles, even if more recent research draws a more complex picture. This aspect of the manuscript fits therefore well into the scope of ACP. In general I agree with Ref 1 remarks about the quantum chemical calculations and this part should be extended.

We would like to thank the referee for reading our manuscript, his valuable comments and overall positive evaluation of our manuscript. Before going to his particular questions, we would like to provide some general comments. First, some of the reviewer's questions and comments concern the theory. Indeed, we try to answer them and incorporate his suggestions in revised manuscript, nevertheless, we would like to stress the point which we have made in our reply to reviewer #1: the major contribution of our present paper is the experiment and the calculations were performed only to provide a support for the unambiguous experimental observations and conclusions. Please, see our reply to reviewer #1, page 1 and 2.

Second, as outlined below, many of the reviewer's questions (points 1-8) originate probably from a slight confusion of our molecular beam method with the condensationchamber type experiments generally used in the aerosol community. Therefore, we provide an extensive explanation of the questioned experimental points for the reviewer below. However, we have not substantially extended the experimental part of our manuscript in this respect, since most of our arguments can be found in basic textbooks on molecular beams and clusters, e.g., [Scoles, G.: Atomic and molecular beam methods, Oxford University Press, 1988; Haberland, H.: Clusters of atoms and molecules. Springer series in chemical physics, vol. 52, Springer-Verlag, 1994; Pauly, H.: Atom, molecule, and cluster beams I & II. Springer-Verlag, 2000]. The conditions specific to our apparatus are then either outlined in the method section or in the cited publications.

Concerning the reviewer's question: "…to my knowledge thermal electrons will most likely attach first to more abundant species. H2SO4 would then react with those ions and form HSO4-…". The reviewer is correct and this has been pointed out in our manuscript on page 2: "*The cosmic rays are the principal source of ionization and free electrons in the upper troposphere and lower stratosphere, where the electron collisions can influence the gas-phase chemistry. Typically, the electrons are effectively thermalized to low energies ($\gtrsim 1$ eV) via multiple inelastic collisions (Campbell and Brunger, 2016) . At these low energies, they are rapidly captured by abundant molecules, in particular $O_2$.*" However, we focus on the mixed $H_2SO_4/H_2O$ clusters and use the electron attachment to ionize these species for the mass spectrometry which delivers some information also about the neutral clusters, their properties and elementary processes in them such as the acidic ionization. In addition, our experiments provide some information about the free electron attachment to these clusters. The main purpose of our study is the molecular level insight into these species and processes, which can be further relevant for atmospheric chemistry.

1) To what temperature does the He/H2SO4/H2O gas mixture drop after nozzle?

It ought to be noted that a supersonic expansion is generally a non-equilibrium process for which the concept of temperature is not well defined. Due to the gas rarefaction in the expansion, the collisions cease at some point (freezing/quitting surface) and the molecules and clusters than fly in the vacuum without further interactions. Typically the molecules undergo $\sim 10^4$ collisions in the expansion and they all happen within ~20 nozzle radii [Pauly, H.: Atom, molecule, and cluster beams I & II. Springer-Verlag, 2000]. In our case this represents ~2 mm from the nozzle throat, and the molecules pass through this region in a few microseconds. Please, see also our answer to referee #1, point I). A temperature can still be used in supersonic expansion to describe the species in the beam, however, one has to be aware that

different degrees of freedom require different number of collisions to cool and equilibrate. Therefore we can speak about rotational, vibrational and translational temperatures which can all be quite different (sometimes even the concept of parallel and perpendicular translational temperature is introduced despite the temperature not being a vector).

Determination of the cluster temperature experimentally is very challenging and has be accomplished only for a few special cases, e.g., for rare gas clusters (~35 K for $Ar_N$) and a few other species by electron diffraction [Farges, J., et al.: Structure and temperature of rare gas clusters in a supersonic expansion. Surf. Sci., 106, 95–100, 1981; Farges, J., et al.: Noncrystalline structure of argon clusters. II. Multilayer icosahedral structure of $Ar_N$ clusters $50 < N < 750$. J. Chem. Phys., 84, 3491–3501, 1986]. For the special case of superfluid helium nanodroplets the vibrational-rotational temperature of a molecule deposited in the cluster was measured spectroscopically giving the temperature of 0.37 K [Hartman, M., et al.: Rotationally resolved spectroscopy of $SF_6$ in liquid helium clusters: A molecular probe of cluster temperature. Phys. Rev. Lett., 75, 1566–1569, 1995]. For small clusters spectroscopic methods can be used in some cases. Nevertheless, for most of the larger molecular clusters, we have to rely on model approaches combined with an indirect experimental evidence to estimate their temperatures, e.g., the evaporative ensemble theory [Klots, C. E.: Evaporative cooling. J. Chem. Phys., 83, 5854–5860, 1985; Klots, C. E.: Temperatures of evaporating cluster. Nature, 327, 222–223, 1987; Klots, C. E.: Kinetic methods for quantifying magic. Z. Phys. D, 21, 335–342, 1991]. A relaxation model was introduced more recently to estimate the temperatures of water clusters [Brudermann, J., et al.: Isomerization and melting-like transition of size-selected water nonamers. J. Phys. Chem. A, 106, 453–457, 2002] and methanol clusters [Steinbach, C., et al.: Isomeric transitions in size-selected methanol hexamers probed by OH-stretch spectroscopy. Phys. Chem. Chem. Phys., 8, 2752–2758, 2006]. Further non-equilibrium numerical models were employed to analyze the cluster temperatures in seeded supersonic expansions of water vapor with Ar and Ne [Jansen, R., et al.: Nonequilibrium numerical model of homogeneous condensation in argon and water vapor expansions. J. Chem. Phys., 132, 244105, 2010; Gimelshein, N., et al.: The temperature and size distribution of large water clusters from a non-equilibrium model. J. Chem. Phys., 142, 244305, 2015], which were also studied spectroscopically by sodium doping method [Buck, U., et al.: A size resolved investigation of large water clusters. Phys. Chem. Chem. Phys., 16, 6859–6871, 2014; Zeuch, T., Buck, U.: Sodium doped hydrogen bonded clusters: Solvated electrons and size selection. Chem. Phys. Lett., 579, 1–10, 2013]. Different models revealed

temperatures between 70 K and 200 K for the water clusters, depending on cluster size and expansion conditions.

It ought to be mentioned that there are special Laval-type nozzle expansions where the equilibrium conditions can be reached, and experiments were designed to study nucleation in the nozzle and post-nozzle flows [Wyslouzil, B. E., et al.: Binary condensation in a supersonic nozzle. J. Chem. Phys., 113, 7317–7329, 2000; Manka, A., et al.: Freezing water in no-man's land. Phys. Chem. Chem. Phys., 14, 4505–4516, 2012; Wyslouzil B. E, Wölk, J.: Overview: Homogeneous nucleation from the vapor phase–The experimental science. J. Chem. Phys. 145, 211702, 2016; Schläppi, B., et al: A pulsed uniform Laval expansion coupled with single photon ionization and mass spectrometric detection for the study of large molecular aggregates. Phys. Chem. Chem. Phys., 17, 25761–25771, 2015; Ferreiro, J. J., et al.: Observation of propane cluster size distributions during nucleation and growth in a Laval expansion. J. Chem. Phys., 145, 211907, 2016; Chakrabarty, S., et al.: Toluene cluster formation in Laval expansions: Nucleation and growth. J. Phys. Chem. A, 121, 3991–4001, 2017]. However, the molecular beam experiments are not well suited to investigate the nucleation of aerosols in general. In molecular beam experiments, the expansion conditions are tuned to produce the desired species and then other experiments are performed with them. We did not attempt to study the nucleation of sulfuric acid but our aim was rather to produce the hydrated sulfuric acid clusters and investigate them by electron attachment and mass spectrometry.

It should be mentioned that the molecular beam experiments can also yield some information useful for molecular level understanding of the nucleation –e.g., in our previous experiments the cross sections of water clusters for pickup of various atmospheric molecules from gas phase could be measured which is essential for the cluster growth [Lengyel, J., et al.: Uptake of atmospheric molecules by ice nanoparticles: Pickup cross sections. J. Chem. Phys., 137, 034304, 2012]. However, this is not the subject of our present publication.

We provide this extensive answer and literature review, to illustrate that the question about the temperature in the expansion is not a trivial one and cannot be easily answered –in fact, this question has been answered so far only for a few specific systems. For the present system, the cluster temperature has not been measured experimentally and the above mentioned theoretical approaches are far beyond the scope of our present research. For pure water clusters the temperature would probably be around 100 K for our present expansion conditions. However, it is well known that already a small admixture of other gas in the expansion can change the conditions significantly, and consequently the resulting

temperature. Therefore we would like to refrain from any speculations about the cluster temperature at this point.

2) What relative humidity would be reached in the cluster formation zone after the nozzle?

Under our most humid conditions (figure 1c in the main paper) the partial water vapor pressure was approximately 42 mbar while the partial vapor pressure of $H_2SO_4$ was 2.04 mbar and we used 2 bar of He buffer gas. Please, see also our reply to Reviewer #1, page 3, last paragraph. RH corresponds to the ratio of the partial pressure of water vapor to the equilibrium vapor pressure of water at the temperature in the expansion. The clusters are formed still in the nozzle within few millimeters from the nozzle throat. The reservoir and nozzle temperatures were 453 K and 458 K, however, these temperatures are irrelevant for the actual temperatures in the gas flow which cools down by the supersonic expansion and the temperature can be as low as 100 K or even lower, leading to supersaturation in the expansion which in turn leads to the cluster formation [Pauly, H.: Atom, molecule, and cluster beams I & II. Springer-Verlag, 2000]. The temperature is not known for the present system, as discussed above, therefore we cannot give any RH value. The partial pressures of component molecules and buffer gas are sufficient to determine the cluster sizes and compositions in supersonic expansions (together with other expansion conditions such as the nozzle temperature, size and shape).

3) How long is the reaction time to form these clusters (before ionisation)?

There are several time-windows in our experiment. First, there is the time during which the collisions occur and the clusters are generated. This has been explained above that the collisions occur within ~2 mm from the nozzle throat, and the molecules pass through this region in about 2 μs (based on the cluster velocity ~1.0-1.5×10$^3$ ms$^{-1}$ which can be actually measured on our apparatus using a pseudo-random chopper and cross-correlation method [Fedor, J., et al.: Cluster cross sections from pickup measurements: Are the established methods consistent? J. Chem. Phys., 135, 104305, 2011; Lengyel, J., et al.: Uptake of atmospheric molecules by ice nanoparticles: Pickup cross sections. J. Chem. Phys., 137, 034304, 2012]). Thus the first "reaction time" in which the clusters are generated is ~2 μs.

Then the collisions cease completely. After the skimmer (~1.5 cm from the nozzle) the beam enters high and ultra-high vacuum regions ($10^{-6}$-$10^{-10}$ mbar) where the mean free path for molecules exceeds several meters, i.e., the clusters fly undisturbed, isolated, not interacting with anything (except for the black-body-radiation photons from the apparatus walls which are at room temperature). The distance to the ionization volume of the TOF

spectrometer is ~1.5 m which the clusters fly in about 1.5 ms. During this time only evaporation of molecules from the cluster can occur (if the clusters were generated in excited states) or spontaneous intra-cluster reactions can happen, e.g., the acidic dissociation. However, we believe that this process occurs already during the cluster generation, as soon as the sulfuric acid molecule is surrounded by enough water molecules in the cluster. So the second "reaction time" of 1.5 ms is rather an idle time. Then the ionization by electron beam occurs.

**4) How long is the reaction time the charged clusters (after ionisation)?**

The electron beam is pulsed for 2 μs during which the clusters can interact with the electrons. After the electron pulse there is a delay of 0.5 μs before the extraction pulse for ions is switched on. During this delay all remaining free electrons leave the ionization volume so that the spectra cannot be disturbed by any effects caused by these electrons accelerated by the extraction voltage. Then the ions are accelerated by an 8 kV pulse into the TOF where the total flight path to the detector is 95 cm. The flight time between the cluster ionization and detection represents several tens of microseconds –the exact value depends on applied voltages, and the cluster mass to charge ratio. The pressure in the TOF is below $10^{-8}$ mbar, i.e., the ions do not collide with any residual gas molecules. Also the cluster density in the beam is so low that any collisions of the nascent cluster ions with other clusters or molecules in the beam are highly unlikely.

In summary, the "reaction time" after the cluster ionization is several tens of microseconds (depending on the mass to charge ratio). However, during this time the cluster ions do not collide with any other species and thus only intracluster reactions can occur. If the cluster ion fragmentation would occur during the time after the cluster extraction and before it enters the reflectron mirror, it would be reflected in the position and shape of the corresponding mass peaks [Wei, S., Castleman, A. W.: Using reflectron time-of-flight mass spectrometer techniques to investigate cluster dynamics and bonding. Int. J. Mass. Spectrom. Ion Processes, 131, 233–264, 1994]. We did not observe any of these effects in the mass spectra, therefore we excluded metastable fragmentation of the clusters after the ionization. Indeed, we observed the cluster ion fragments in the mass spectra, but the fragmentation had to be fast, completed between the ionization and extraction pulse, i.e. within 0.5 μs (taking into account also the ionization pulse duration, the maximum time available for this fragmentation could be 2.5 μs).

**5) Is it possible that some water is condensing on the cluster after the ionisation?**

The cluster formation, which includes also the addition of water molecule, is strictly connected only to the supersonic expansion. As described above, the TOF chamber is ~1.5 m downstream from the nozzle chamber divided by several differentially pumped vacuum chambers and there are no further collisions possible in the ultrahigh vacuum in our TOF chamber (the corresponding pressure of ~$10^{-8}$ mbar corresponds to the mean free path in the order of 10 km for molecules).

**6) Could you determine the lifetime of the intermediate by adjusting the reaction time?**

We are not sure what the Reviewer means by the "intermediate" and "reaction time" but we have probably answered this question above, point 4). If some intermediate is generated after the electron attachment which then fragments to the final cluster ion, the fragmentation has to occur within the 0.5 μs delay between the electron pulse and the ion extraction (at most 2.5 μs if we take into account also the ionization pulse duration). Any metastable decay which would occur after the ion extraction during the flight time to the reflectron mirror would be recognized in the spectral peak position and shape. Since no such effects were observed in the spectra the intermediate lifetime must be shorter than 0.5 μs. We could make this delay time even shorter, however, some unwanted effects from residual free electrons could disturb the spectrum. Therefore this is the shortest time window we could provide at the moment.

**7) I assume the charged clusters are accelerated by an electric field in a low pressure environment, is evaporation and fragmentation a potential problem in your setup?**

We believe that we have answered this question already above. If the cluster mass changes after its extraction during its flight time to the reflectron mirror, the fingerprint of the mass change can be unambiguously recognized in the spectrum [Wei, S., Castleman, A. W.: Using reflectron time-of-flight mass spectrometer techniques to investigate cluster dynamics and bonding. Int. J. Mass. Spectrom. Ion Processes, 131, 233–264, 1994]. There was no evidence for such metastable fragmentation in our spectra.

**8) How do your dissociation constants compare to bulk phase sulphuric acid?**

The dissociation constant is a characteristics of a macroscopic bulk systems in equilibrium. For individual isolated clusters in vacuum we cannot measure the dissociation constants in our experiment. From our mass spectra we can make some conclusions about the acidic dissociation of sulfuric acid in water clusters. In principle we can answer the question: how many water molecules are needed to acidically dissociate a sulfuric acid molecule? Actually,

this question has been treated by similar methods in the past for nitric acid [Kay, B. D., et al.: Studies of gas-phase clusters: The solvation of $HNO_3$ in microscopic aqueous clusters, Chem. Phys. Lett., 80, 469, 1981] (see also our answer to point 3) of reviewer #1. Such question cannot be solved in the bulk where the dissociation constant characterizes the macroscopic system. However, it is a fundamental question and the answer to this question provides molecular level understanding to the processes occurring in the bulk.

9) Data Statement is missing. https://www.atmospheric-chemistry-and-physics.net/about/data_policy.html

The data statement has been added at the end of the manuscript.

10) A brief overview of current developments in the quantum chemical treatment of H2SO4-H2O clusters is missing.

As mentioned above, the major contribution of our present paper is the unique experiment. The calculations were performed to provide a support for the experimental conclusions. The major conclusions, e.g., about the acidic dissociation or fragment caging in the clusters could be derived essentially just based on the experimental evidence. Please, see our reply to the reviewer #1, page 1-2 for more details. Besides, we would like to point out that our benchmark calculations proved that the used computational approach is in reasonable agreement with the higher-level *ab initio* methods. These benchmark calculations are now presented in supporting information and also discussed in our reply to the reviewer #1 on p. 2.

Since the present manuscript is not a theoretical paper, and as experimentalists we rather focus on the previous experimental evidence directly relevant to the investigated phenomena, we believe that an extensive overview of the previous quantum chemical calculations of the $H_2SO_4$-$H_2O$ system would unnecessarily extend the theory-related part of our manuscript and distract the reader from the main message. We believed that we have covered most of the recent theoretical papers directly relevant to our investigations in our references. However, we could miss some important contributions and we would be happy to add some more references, if the reviewer provides some explicit suggestions what we missed in our survey.

11) P 1 L 19: I think aerosols should be replaced with clusters as the paper shows only results for clusters containing a few sulphuric acid molecules

We followed the reviewer's suggestion and replace "aerosols" with "clusters".

12) P 1 L 27: I suggest to replace neutral structure by undissociated structure and (optionally) ion pair structure by dissociated structure (ps: probably solved by following referee 1 request)

We followed the suggestion of both referees and the respective terminology has been changed to avoid any confusion.

13) P2 L 3-9: Is the Sulphuric acid charging by e- capture actually important for the production of charged sulphuric acid cluster (especially in the context of boundary layer nucleation events)? Some older and more recent literature is missing in the context of ion induced nucleation Raes and Jansen 1985 DOI: 10.1016/0021-8502(85)90028-X (or even earlier), a more recent one would be Kirkby et al 2011 (DOI: 10.1038/nature10343) or the most recent for H2SO4 water Kürten et al 2016 (10.1002/2015JD023908). There are also field observations that try to estimate the fraction of ion induced nucleation in the total nucleation rate (Hirsiko et al 2011, doi:10.5194/acp-11-767-2011).

As already mentioned above, we do not focus on the nucleation of atmospheric sulfuric acid aerosols. We investigate the properties of the individual mixed $H_2SO_4$-$H_2O$ clusters which have not been observed by mass spectrometric experiments before. Therefore we have avoided an extensive discussion of the atmospheric nucleation of sulfuric acid aerosols in the introduction, since we found it unnecessarily lengthening our paper and distracting the reader from the main topic. Nevertheless, we have now added a brief overview of the nucleation in the introduction on page 2 citing the above publications:

14) P2 L14-15: Charged sulphuric acid clusters in the atmosphere where already measured before, example literature Eisele, 1989 DOI: 10.1029/JD094iD02p02183 I have the impression this sentence is out of context and should be moved to the previous paragraph.

The introduction has been changed as outlined above.

15) Methods: P4 L 2: What was the criterion for choosing the clusters for reoptimisation? How many isomers were used?

We used previously observed energetic minima from literature as the initial structures and equilibrated them in molecular dynamics runs. From the MD simulation, several structures were randomly taken and re-optimized at the M06-2X/aug-cc-pVDZ level. Altogether 8 different isomers on average were optimized from various starting structures for each cluster type $((H_2SO_4)_{1-2}(H_2O)_{0-5}$, $(H_2SO_4)_{0-1}(H_2O)_{0-5}HSO_4^-)$ including the hydrogen-bonded, $H_2SO_4\cdots H_2O$, and ion-pair, $HSO_4^-\cdots H_3O^+$, structures in neutral clusters. Only the most stable

isomers were considered for further calculations. The section '*2 Experimental and theoretical methods*' has been extended with the aforementioned discussion.

16) Results: P4 L 17: The results of the quantum chemical calculations should be compared to previous work.

The comparison with the previous theoretical work has been added to the discussion concerning the binary nucleation of $H_2SO_4/H_2O$ clusters. As an example, the calculated free energies of the addition of water molecule were in reasonable agreement with the literature values [Kurtén, T., et al.: Quantum chemical studies of hydrate formation of $H_2SO_4$ and $HSO_4^-$. Boreal Environ. Res., 12, 431, 2007; Loukonen, V., et al.: Enhancing effect of dimethylamine in sulfuric acid nucleation. Atmos. Chem. Phys., 10, 4961, 2010; Henschel, H., et al.: Hydration of atmospherically relevant molecular clusters: Computational chemistry and classical thermodynamics. J. Phys. Chem. A, 118, 2599, 2014]. The seemingly different values in free energies were caused by different definitions of the nucleation event in the calculated chemical reactions:

$$H_2SO_4(H_2O)_{n-1} + H_2O \rightarrow H_2SO_4(H_2O)_n \qquad \Delta_r G^0(1)$$

$$H_2SO_4 + nH_2O \rightarrow H_2SO_4(H_2O)_n \qquad \Delta_r G^0(2)$$

In the present paper, we calculate the free energy $\Delta_r G^0(1)$ for an addition of single water molecule to a preexisting cluster. The free energy can be calculated also from the accumulation of all molecular component to the cluster $\Delta_r G^0(2)$. The former way of calculation, $\Delta_r G^0(1)$, is more illustrative for our discussion. When we calculated $\Delta_r G^0(2)$, it was in a good agreement with the previously published data as shown in Table 1.

**Table 1**: Free energies (in kcal mol$^{-1}$, at $T$=298K and $p^0$=1atm) of binary nucleation of $H_2O/H_2SO_4$ clusters

| $n =$ | $\Delta_r G^0(1)$ our results | $\Delta_r G^0(1)$ Kurtén, et al. | $\Delta_r G^0(2)$ our results | $\Delta_r G^0(2)$ Henschel, et al. | $\Delta_r G^0(2)$ Loukonen, et al. |
|---|---|---|---|---|---|
| 1 | −2.7 | −2.81 | −2.7 | −2.60 | −2.93 |
| 2 | −1.7 | −1.87 | −4.4 | −4.40 | −6.26 |
| 3 | −0.8 | −2.37 | −5.2 | −5.83 | −7.11 |
| 4 | −2.3 | −0.90 | −7.5 | −7.05 | −8.11 |
| 5 | −1.2 | − | −8.7 | −6.81 | −10.01 |

The corresponding discussion has been added to the manuscript, and the respective chemical equations included in a new Table 1 in the manuscript.

**17) P8 L 21: could this be quantified more? Is the cross section and e- concentration high enough so that electron attachment in the troposphere is a major source of HSO4-? What is the free e- concentration in the troposphere?**

The cosmic ray ionization rate varies between about 2 ion pairs $cm^{-3}$ $s^{-1}$ close to Earth's surface and 40 ion pairs $cm^{-3}$ $s^{-1}$ at the top of the troposphere [Carslaw, K. S., et al.: Cosmic rays, clouds, and climate, Science, 298, 1732–1737, 2002]. The observed density of free electrons is very low in stratosphere around $5\times10^3$ $cm^{-3}$ [Smith, D., Adams, N. G.: Elementary plasma reactions of environmental interests. Top. Curr. Chem. 89, 1–43, 1980] because they rapidly interact with abundant molecules, in particular $O_2$. For the lower altitudes, the number of the free electrons is significantly reduced. Therefore we expect a minor importance of free electrons for tropospheric ion chemistry. The tropospheric $HSO_4^-$ ions are most likely formed in reaction of gas-phase sulfuric acid and various molecular anions, such as $O_2^-$. At higher altitudes the contribution of free electrons is larger, therefore we have suggested that the free electron attachment may actually contribute. However, our major focus is the molecular level understanding to the mixed $H_2SO_4/H_2O$ clusters and the electron attachment process to them. We are not qualified to make speculations about the actual contribution of such processes, however, the corresponding sentence can stimulate the interest of scientists modelling the atmospheric processes.

**18) P9 L 3: How was that calculated? You probably used all structures for a given isomer, please be a bit more precise here.**

The computational procedure was described in detail in the section '*2 Experimental and theoretical methods*', page 4, line 3 (in the original manuscript): "Only the most stable isomers were considered for further calculations."

**19) Conclusion: P10 L 16-17: The dipole in the dissociated cluster could also enhance the uptake of ions such as NO3-. With the quantum chemical calculations you could calculate this enhancement (see Su & Chesnavich DOI:10.1063/1.442828).**

The reviewer is correct and the dipole or positive charge in the dissociated cluster could also enhance the uptake of ions such as $NO_3^-$. This could be calculated theoretically. However, we would like to stress again, that the main value of our contribution is the solid experimental evidence from an advanced experiment with particles in the molecular beam. To perform such

experiment for the proposed $NO_3^-$ ion attachment would require substantial changes in our experimental arrangement and would be difficult –for example, already the initial question of a good source for an intense $NO_3^-$ ion beam would be difficult to solve. This goes far beyond not even our present publication but also our near future experimental plans. But we thank the reviewer for an interesting input for some future plans.

Our present study is focused on the binary $H_2SO_4/H_2O$ clusters. Apparently the detection of these hydrated clusters seems to be not trivial for laboratory experiments since they have not been detected in the aerosol chamber experiments so far [Kirkby, J., et al.: Role of sulphuric acid, ammonia and galactic cosmic rays in atmospheric aerosol nucleation, Nature, 476, 429, 2011, Kürten, A., et al.: Neutral molecular cluster formation of sulfuric acid-dimethylamine observed in real time under atmospheric conditions. Proc. Natl. Acad. Sci. USA, 111, 15019, 2014]. Here we have demonstrated an efficient way of generating these hydrated sulfuric acid clusters using the molecular beam technique, and characterized them by the electron attachment mass spectrometry. Adding additional calculations of ternary complexes with $NO_3^-$ ion to our present manuscript would be confusing without any experiment connected to this topic.

20) figure 2: The figure and data will be much easier to comprehend if the dotted lines are replaced by symbols (e.g. square, disk, triangle up/down) at the top of the corresponding peaks. currently the colours are too similar.

The symbols have been added to Figure 2.

21) figure 3: You should provide the geometry of the clusters as text files (pdb or xyz). Allows readers to visualise the results in a molecule viewer.

The respective geometries have been added to the Supplement.

22) figure 4: If possible combine the curves in figure 4 for each row, using different colours and symbols

We agree with the reviewer that there seems not to be much information in figure 4, except that the energy dependent ion yields look essentially all the same. However, that is exactly our message. As outlined also in the text, the ion yield curves were identical for all the mass peaks (within the experimental errors). Figure 4 should illustrate this point for a few examples (there are plenty of electron energy dependencies measured for other mass peaks, and we have added some more examples in the supporting information for illustration). If we plotted all the energy dependencies in one single plot (normalized) they would be just overlapping. We

believe that showing the individual plots separately provides a clearer picture, and the reader can also see the different signal intensities and the level of the noise in the data. This presentation of figure 4 provides a clear picture of what is meant in the text by "the same energy dependencies". We believe that this message would not be so clear in the representation suggested by the reviewer, therefore we would like to keep the present form of figure 4.

[revised manuscript text omitted]

**S1 Experimental setup**

The experiments were performed on a versatile and unique experimental apparatus CLUB (cluster beam apparatus) which allows a variety of different experiments with a molecular beam of isolated clusters in vacuum. The apparatus and experiments have been described in numerous publications previously (e.g., mass spectrometry: (Lengyel et al., 2012; Kočišek et al., 2013a; Kočišek et al., 2013b); electron attachment: (Kočišek et al., 2016a; Kočišek et al., 2016b; Lengyel et al., 2016); etc.) and the details can be found in these references. The sketch of the CLUB apparatus is shown in Fig. S1 below. In the present work, the clusters were produced in the first vacuum chamber by supersonic expansion of the sulfuric acid vapor with buffer gas He, i.e. a mixture of $H_2SO_4$, $H_2O$ and He gas phase molecules. The present mass spectrometry was performed in the $4^{th}$ vacuum chamber TOFMS, where the cluster beam was crossed by a low-energy electron beam. Further details are given in the experimental section of the present paper. The other options and features of the CLUB apparatus shown in Fig. S1 were not exploited in the present experiments.

[Figure]

**Figure S1: Schematic overview of the CLUB apparatus: VMI –velocity map imaging for photodissociation of molecules in clusters; TOFMS –reflectron time-of-flight mass spectrometer with various ionization methods, e.g., electron ionization, electron attachment, photoionization; QMS –quadrupole mass spectrometer with electron ionization.**

**S2 Dipole moment of H₂SO₄(H₂O)₅ clusters**

Our M06-2X/aug-cc-pVDZ calculations exhibit only a small change in the cluster dipole moment upon the acidic dissociation of the sulfuric acid molecule on water cluster and could be overlapped by dynamic effects. The calculated dipole moments summarized in Figure S2 were in very broad range from ~0.3 D to ~4.6 D for H₂SO₄(H₂O)₅ clusters which depend rather on the cluster structure than on the acidic dissociation. Most likely, not only the energy minimum structure (see Figure S2 (a) for ion-pair and (e) for covalently-bonded H₂SO₄) but many different cluster structures are generated in the supersonic expansion.

[Figure]

**Figure S2: Selected local minima of neutral, H₂SO₄···H₂O, (a-d) and ion-pair, HSO₄⁻···H₃O⁺, (e-h) structures in H₂SO₄(H₂O)₅ clusters and the corresponding dipole moments calculated at the M06-2X/aug-cc-pVDZ level of theory.**

**S3 Thermochemistry**

**Table S1: Reaction energies (in kJ mol$^{-1}$) for the HSO$_4^-$ dissociation channels after electron attachment to H$_2$SO$_4$/H$_2$O clusters optimized at the M06-2X/aug-cc-pVDZ level of theory.**

| $N$ | H$_2$SO$_4$(H$_2$O)$_N$ | (H$_2$SO$_4$)$_2$(H$_2$O)$_N$ |
|---|---|---|
| 0 | −8.6 | −132.2 |
| 1 | −16.2 | −117.2 |
| 2 | −30.3 | −120.5 |
| 3 | −39.6 | −99.8 |
| 4 | −43.5 | −106.5 |
| 5 | −46.3 | −88.9 |

**Table S2: Free energies (in kJ mol$^{-1}$, at $T$=298K and $p^0$=1atm) of binary nucleation of H$_2$O/H$_2$SO$_4$ clusters**

H$_2$SO$_4$(H$_2$O)$_{n-1}$ + H$_2$O → H$_2$SO$_4$(H$_2$O)$_n$ $\quad\quad\quad\quad$ $\Delta_rG^0(1)$

H$_2$SO$_4$ + $n$H$_2$O → H$_2$SO$_4$(H$_2$O)$_n$ $\quad\quad\quad\quad$ $\Delta_rG^0(2)$

| $n$ = | $\Delta_rG^0(1)$ our results | $\Delta_rG^0(1)$ (Kurtén et al., 2007) | $\Delta_rG^0(2)$ our results | $\Delta_rG^0(2)$ (Henschel et al., 2014) | $\Delta_rG^0(2)$ (Loukonen et al., 2010) |
|---|---|---|---|---|---|
| 1 | −11.3 | −11.76 | −11.3 | −10.88 | −12.26 |
| 2 | −7.1 | −7.82 | −18.4 | −18.41 | −26.19 |
| 3 | −3.3 | −9.92 | −21.8 | −24.39 | −29.75 |
| 4 | −9.6 | −3.77 | −31.4 | −29.50 | −33.93 |
| 5 | −5.0 | − | −36.4 | −28.49 | −41.88 |

[Figure]

**Figure S3: Ion-yield curves for selected ionic fragments with different degree of hydration.**

**S5 Benchmarking the electronic structure calculations**

Table 1 summarizes the benchmark calculations of electron affinity of $HSO_4$, ionization potential of $H_2SO_4$, and reaction enthalpies for deprotonation of gas-phase $H_2SO_4$ calculated at different levels of theory. The M06-2X/aug-cc-pVDZ energies are comparable with the CCSD/aug-cc-pVDZ values with the exception of the $IP(H_2SO_4)$. The comparison of double-zeta with triple-zeta basis sets of the M06-2X functional shows that there is essentially constant shift from the experimental values and therefore we do not expect any significant shift in reaction energies even upon hydration. The calculated reaction enthalpies for deprotonation of gas-phase $H_2SO_4$ are in good agreement with the experimental value. The error of the DFT method is 0.1-0.2 eV. Please note that, in the present work, chemical trends with respect to hydration are of the main concern, and a possible systematic shift of few tenths of eV does not influence our conclusions.

**Table S3: Electron affinity of $HSO_4$, ionization potential of $H_2SO_4$, and enthalpy of deprotonation at various levels of theory (all in kJ mol$^{-1}$). DZ and TZ represent aug-cc-pVDZ and aug-cc-pVTZ, respectively. Enthalpies were calculated at 298.15 K within the harmonic approximation.**

| | B3LYP/DZ | M06-2X/DZ | M06-2X/TZ | MP2/DZ | CCSD/DZ | Experiment |
|---|---|---|---|---|---|---|
| $EA(HSO_4)$ | 453 | 474 | 483 | 503 | 478 | 458±10 (Wang et al., 2000) |
| $IP(H_2SO_4)$ | 1103 | 1117 | 1137 | 1195 | 1209 | 1196±5 (Snow and Thomas, 1990) |
| $\Delta H(H_2SO_4 \rightarrow H^+ + HSO_4^-)$ | 1318 | 1304 | 1300 | 1294 | 1309 | 1295±23 (Wang et al., 2000) |

**Structures optimized at the M06-2X/aug-cc-pVDZ level of theory (coordinates in Å)**

H2SO4,

| | | | |
|---|---|---|---|
| S | 0.000064 | -0.000011 | 0.150574 |
| O | -0.008803 | 1.287414 | 0.823392 |
| O | 0.008537 | -1.287472 | 0.823344 |
| O | -1.253357 | -0.047337 | -0.873079 |
| O | 1.253426 | 0.047310 | -0.873222 |
| H | -1.477412 | 0.867560 | -1.107276 |
| H | 1.477544 | -0.867564 | -1.107433 |

H2SO4.H2O,

| | | | |
|---|---|---|---|
| S | -0.002092 | 0.478526 | 0.019730 |
| O | -1.122963 | 1.157172 | -0.631673 |
| O | 1.114572 | 1.190582 | 0.618755 |
| O | -0.562309 | -0.538977 | 1.107359 |
| O | 0.584738 | -0.519520 | -1.120247 |
| H | -1.483379 | -0.811275 | 0.821010 |
| H | 1.486119 | -0.765134 | -0.858585 |
| O | -2.970064 | -0.852470 | 0.101511 |
| H | -3.213990 | -1.565378 | -0.497780 |
| H | -2.837433 | -0.066926 | -0.451981 |

H2SO4.2H2O,

| | | | |
|---|---|---|---|
| S | 0.000000 | 0.467392 | 0.000000 |
| O | -1.127912 | 1.182899 | -0.598603 |
| O | 1.127912 | 1.182899 | 0.598603 |
| O | -0.544571 | -0.525972 | 1.126354 |
| O | 0.544571 | -0.525972 | -1.126354 |
| H | -1.457398 | -0.819564 | 0.845467 |
| H | 1.457398 | -0.819564 | -0.845467 |
| O | -2.952264 | -0.863317 | 0.087315 |
| H | -3.167318 | -1.556293 | -0.545456 |
| H | -2.804014 | -0.059401 | -0.435852 |
| O | 2.952264 | -0.863317 | -0.087315 |
| H | 3.167318 | -1.556293 | 0.545456 |
| H | 2.804014 | -0.059401 | 0.435852 |

H2SO4.3H2O,

| | | | |
|---|---|---|---|
| O | 1.534809 | 1.462137 | -1.001570 |
| H | 1.781577 | 0.721103 | -1.570209 |
| H | 2.010710 | 1.291324 | -0.171101 |
| S | -1.254013 | -0.027940 | 0.145515 |
| O | -2.676737 | -0.212944 | 0.334029 |
| O | -1.037139 | 1.389948 | -0.533069 |
| O | -0.755988 | -1.052734 | -0.979924 |
| O | -0.331366 | -0.161830 | 1.295309 |
| H | -0.055388 | 1.501559 | -0.728110 |
| H | 0.213448 | -1.230135 | -0.856840 |
| O | 1.907043 | -1.412438 | -0.785868 |
| H | 2.255817 | -1.097554 | 0.069469 |
| H | 2.246433 | -2.302996 | -0.917273 |
| O | 2.326583 | 0.197862 | 1.379342 |
| H | 2.786076 | 0.330487 | 2.212974 |
| H | 1.369934 | 0.132151 | 1.580426 |

H2SO4.4H2O,

| | | | |
|---|---|---|---|
| O | 0.369118 | 1.579893 | -0.259299 |
| H | 1.202090 | 1.605634 | 0.346984 |
| S | 0.347838 | 0.202507 | -1.049951 |
| O | 1.673499 | -0.093974 | -1.603650 |
| O | 0.109356 | -0.873963 | 0.093257 |
| H | -0.835252 | -0.743440 | 0.484336 |
| O | -0.815844 | 0.274657 | -1.940218 |
| O | 3.128602 | -1.008646 | 0.656279 |
| H | 2.514904 | -1.630884 | 1.062595 |
| H | 2.800389 | -0.920750 | -0.255144 |
| O | -2.695346 | 1.776866 | -0.432679 |
| H | -2.179271 | 1.430809 | -1.181062 |

| | | | |
|---|---|---|---|
| H | -2.166168 | 2.514883 | -0.109840 |
| O | 2.413802 | 1.506984 | 1.243770 |
| H | 2.769558 | 0.589206 | 1.141609 |
| H | 3.139971 | 2.102662 | 1.033068 |
| O | -2.207302 | -0.418736 | 1.025772 |
| H | -2.499986 | 0.413818 | 0.577955 |
| H | -2.891058 | -1.073226 | 0.850620 |

H2SO4.5H2O,

| | | | |
|---|---|---|---|
| O | 2.530094 | -0.488264 | 0.456179 |
| O | 2.766862 | 1.149864 | -1.439462 |
| O | 1.747942 | -2.556670 | -1.273177 |
| O | -0.058200 | 0.544141 | 0.545237 |
| S | -0.833135 | -0.036079 | -0.582157 |
| O | -0.673560 | 0.720459 | -1.870133 |
| O | -0.672560 | -1.504548 | -0.771780 |
| O | -2.372431 | 0.198128 | -0.113351 |
| O | 0.908546 | 2.840738 | -0.960190 |
| H | 1.624235 | -0.272759 | 0.750816 |
| H | 2.451809 | -1.361197 | 0.030348 |
| H | 1.818091 | -3.515374 | -1.300637 |
| H | 0.205597 | 2.354613 | -1.433529 |
| H | 0.670757 | 2.675538 | -0.036384 |
| H | 0.813063 | -2.344304 | -1.056790 |
| H | -2.948688 | -0.173522 | -0.798367 |
| H | 2.094086 | 1.898372 | -1.246379 |
| H | 2.755855 | 0.515316 | -0.617185 |
| H | 2.352817 | 0.566049 | -2.187776 |
| H | 1.724621 | -1.281620 | -2.713774 |
| O | 1.610395 | -0.384716 | -3.074888 |
| H | 0.679306 | -0.153067 | -2.895336 |

2H2SO4,

| | | | |
|---|---|---|---|
| S | 1.745825 | 0.029839 | -0.059980 |
| O | 3.329326 | 0.099758 | -0.237040 |
| O | 1.225540 | 1.341119 | -0.440342 |
| O | 1.681857 | -0.172605 | 1.515730 |
| O | 1.235030 | -1.168892 | -0.740880 |
| H | 3.685237 | -0.804701 | -0.247897 |
| H | 0.719563 | -0.220633 | 1.736599 |
| S | -1.924079 | -0.019868 | 0.098364 |
| O | -1.563209 | -1.143575 | -0.981219 |
| O | -3.351604 | -0.059240 | 0.306312 |
| O | -1.584903 | 1.348763 | -0.655891 |
| O | -0.981346 | -0.160287 | 1.227890 |
| H | -0.592141 | -1.293207 | -0.969850 |
| H | -0.616274 | 1.506078 | -0.611788 |

2H2SO4.1H2O,

| | | | |
|---|---|---|---|
| S | -1.353553 | -0.511414 | -0.087013 |
| O | -1.227393 | -0.561068 | 1.499813 |
| O | -1.128619 | 0.883270 | -0.538941 |
| O | -2.864381 | -0.884391 | -0.260993 |
| O | -0.530691 | -1.558253 | -0.696590 |
| H | -0.279978 | -0.347271 | 1.693110 |
| H | -3.405755 | -0.031417 | -0.155546 |
| S | 2.251621 | 0.254920 | 0.108457 |
| O | 2.223385 | -1.111167 | -0.718891 |
| O | 1.345124 | 0.113600 | 1.267819 |
| O | 1.594297 | 1.312963 | -0.894274 |
| O | 3.636755 | 0.618520 | 0.292603 |
| H | 1.294006 | -1.430291 | -0.763406 |
| H | 0.613364 | 1.264162 | -0.813656 |
| O | -3.918220 | 1.453703 | -0.075381 |
| H | -3.105060 | 1.963084 | -0.204107 |
| H | -4.331662 | 1.791292 | 0.726718 |

2H2SO4.2H2O,

| | | | |
|---|---|---|---|
| S | 1.133014 | -0.781556 | -0.436442 |
| O | 1.107072 | -0.548213 | 1.139465 |
| O | 2.498648 | -1.195467 | -0.751781 |
| O | 0.831395 | 0.660660 | -1.022179 |
| O | 0.001521 | -1.622536 | -0.845083 |
| H | 0.188558 | -0.231213 | 1.368849 |
| H | 1.648914 | 1.315414 | -0.812327 |
| S | -2.450541 | 0.296644 | 0.340762 |
| O | -1.979223 | 1.026965 | -1.004861 |
| O | -1.338323 | 0.400646 | 1.313497 |
| O | -2.605880 | -1.226151 | -0.102257 |
| O | -3.764970 | 0.783930 | 0.683121 |
| H | -1.000647 | 0.985115 | -1.071389 |
| H | -1.718447 | -1.558502 | -0.372102 |
| O | 4.165046 | 0.366943 | 0.940214 |
| H | 3.793831 | -0.421900 | 0.511638 |
| H | 3.970992 | 0.264946 | 1.878140 |
| O | 2.749942 | 2.106272 | -0.451276 |
| H | 3.281491 | 2.434290 | -1.184733 |
| H | 3.352697 | 1.559276 | 0.121564 |

2H2SO4.3H2O,

| | | | |
|---|---|---|---|
| S | 1.911406 | 0.302172 | 1.004299 |
| O | 2.002457 | -0.519031 | -0.354128 |
| O | 3.251986 | 0.516643 | 1.497074 |
| O | 1.163196 | -0.684751 | 2.002934 |
| O | 1.013037 | 1.465973 | 0.766709 |
| H | 1.082924 | -0.652053 | -0.723374 |
| H | 0.231275 | -0.829764 | 1.678738 |
| S | -1.640653 | -0.938902 | -0.382259 |
| O | -0.465014 | -0.686075 | -1.281778 |
| O | -2.024913 | -2.497877 | -0.553992 |
| O | -2.837837 | -0.154923 | -0.791105 |
| O | -1.312562 | -0.843515 | 1.068651 |
| H | -2.305329 | -2.646579 | -1.470978 |
| O | -2.325283 | 2.390213 | -0.373368 |
| H | -2.018102 | 2.289516 | 0.620738 |
| H | -1.448985 | 2.403053 | -0.944100 |
| H | -2.714090 | 1.496154 | -0.584884 |
| O | -0.187319 | 2.160939 | -1.662509 |
| H | 0.493477 | 2.109950 | -0.966892 |
| H | -0.257542 | 1.239238 | -1.962555 |
| O | -1.484569 | 1.913077 | 1.941515 |
| H | -1.722947 | 0.978659 | 2.051025 |
| H | -0.517672 | 1.880340 | 1.817961 |

2H2SO4.4H2O,

| | | | |
|---|---|---|---|
| S | 1.784275 | 0.547719 | -0.487643 |
| O | 0.376649 | 1.048498 | -0.879104 |
| O | -0.981482 | -1.044910 | -1.422991 |
| H | -0.520105 | -0.142336 | -1.325710 |
| O | 2.488009 | -0.033265 | -1.644329 |
| O | 1.504115 | -0.718557 | 0.461352 |
| O | 1.067604 | -2.379082 | -2.044191 |
| H | -0.208564 | -1.691632 | -1.666255 |
| O | -0.628815 | 2.820358 | 0.642983 |
| H | -1.347252 | 2.233968 | 1.056338 |
| S | -1.585857 | -0.132154 | 1.888293 |
| O | -0.168479 | 0.160378 | 2.261477 |
| O | -1.715621 | -1.338022 | 1.009703 |
| O | -2.321114 | 1.068096 | 1.404662 |
| H | 0.921306 | -0.429200 | 1.224166 |
| O | 2.489556 | 1.589776 | 0.284729 |
| H | 1.374637 | -2.963366 | -1.340778 |
| H | 1.718500 | -1.643959 | -2.049245 |
| H | -1.320159 | -1.237200 | -0.482771 |
| H | -0.185312 | 2.196170 | -0.033581 |
| H | 0.100541 | 3.046060 | 1.335600 |

| | | | |
|---|---|---|---|
| O | -2.369059 | -0.507824 | 3.251984 |
| H | -2.013358 | -1.347082 | 3.583243 |
| H | 1.775657 | 4.057102 | 2.168376 |
| O | 1.340476 | 3.198028 | 2.166567 |
| H | 1.931806 | 2.593921 | 1.671787 |

2H2SO4.5H2O,

| | | | |
|---|---|---|---|
| S | 2.463316 | -0.007738 | -0.264464 |
| O | 1.490393 | -0.370175 | 0.797581 |
| O | 3.883207 | -0.484662 | 0.361638 |
| O | 2.600433 | 1.455901 | -0.521683 |
| O | 2.285757 | -0.790802 | -1.528680 |
| H | 4.582422 | -0.267420 | -0.274512 |
| S | -2.346417 | -0.293625 | 0.162067 |
| O | -1.629343 | -0.359861 | -1.151363 |
| O | -3.794822 | -0.921159 | -0.223972 |
| O | -2.596308 | 1.106970 | 0.615838 |
| O | -1.768006 | -1.142111 | 1.232780 |
| H | -4.357598 | -0.882783 | 0.564910 |
| O | 0.165337 | -2.193356 | -1.420715 |
| H | 1.026470 | -1.660524 | -1.541397 |
| H | 0.229414 | -2.605931 | -0.459033 |
| H | -0.579199 | -1.512094 | -1.369709 |
| O | 0.612327 | 2.885592 | 0.249474 |
| H | 1.428177 | 2.386092 | -0.074238 |
| H | 0.269836 | 2.362126 | 1.089286 |
| H | -0.121548 | 2.805885 | -0.467326 |
| O | -0.201685 | 1.490133 | 2.156993 |
| H | -1.136652 | 1.294638 | 1.972911 |
| H | 0.271242 | 0.659281 | 1.979821 |
| O | -1.311766 | 2.570989 | -1.373026 |
| H | -1.984887 | 2.240590 | -0.746780 |
| H | -1.170761 | 1.803888 | -1.945146 |
| O | 0.302146 | -2.945633 | 0.961722 |
| H | -0.519241 | -2.509320 | 1.258325 |
| H | 0.989487 | -2.323997 | 1.251641 |